# Characterization of Tellurite Toxicity to *Escherichia coli* Under Aerobic and Anaerobic Conditions

**DOI:** 10.3390/ijms26157287

**Published:** 2025-07-28

**Authors:** Roberto Luraschi, Claudia Muñoz-Villagrán, Fabián A. Cornejo, Benoit Pugin, Fernanda Contreras Tobar, Juan Marcelo Sandoval, Jaime Andrés Rivas-Pardo, Carlos Vera, Felipe Arenas

**Affiliations:** 1Laboratorio de Microbiología Molecular, Facultad de Química y Biología, Universidad de Santiago de Chile, Santiago 9170022, Chile; r.luraschiv@gmail.com (R.L.); claudia.munoz.v@usach.cl (C.M.-V.); fabian.cornejo@usach.cl (F.A.C.); carlos.vera.v@usach.cl (C.V.); 2Laboratory of Food Biotechnology, Department of Health Sciences and Technology, ETH Zürich, 8092 Zürich, Switzerland; benoit.pugin@hest.ethz.ch; 3Laboratorio de Ecología de Ambientes Extremos, Centro de Genómica, Ecología y Medio Ambiente (GEMA), Universidad Mayor, Santiago 8580745, Chile; fernanda.contreras.t@usach.cl; 4Facultad de Ciencias, Universidad Arturo Prat, Iquique 1110939, Chile; jusandoval@unap.cl; 5Laboratorio de Genómica Microbiana, Escuela de Biotecnología, Centro de Genómica y Bioinformática, Facultad de Ciencias, Ingeniería y Tecnología, Universidad Mayor, Santiago 8580745, Chile; jaime.rivas@umayor.cl

**Keywords:** tellurite, anaerobiosis, aerobiosis, *E. coli*, chemical genomics, amino acids, plasma membrane fluidity, metabolomics

## Abstract

Tellurite (TeO_3_^2−^) is a highly soluble and toxic oxyanion that inhibits the growth of *Escherichia coli* at concentrations as low as ~1 µg/mL. This toxicity has been primarily attributed to the generation of reactive oxygen species (ROS) during its intracellular reduction by thiol-containing molecules and NAD(P)H-dependent enzymes. However, under anaerobic conditions, *E. coli* exhibits significantly increased tellurite tolerance—up to 100-fold in minimal media—suggesting the involvement of additional, ROS-independent mechanisms. In this study, we combined chemical-genomic screening, untargeted metabolomics, and targeted biochemical assays to investigate the effects of tellurite under both aerobic and anaerobic conditions. Our findings reveal that tellurite perturbs amino acid and nucleotide metabolism, leading to intracellular imbalances that impair protein synthesis. Additionally, tellurite induces notable changes in membrane lipid composition, particularly in phosphatidylethanolamine derivatives, which may influence biophysical properties of the membrane, such as fluidity or curvature. This membrane remodeling could contribute to the increased resistance observed under anaerobic conditions, although direct evidence of altered membrane fluidity remains to be established. Overall, these results demonstrate that tellurite toxicity extends beyond oxidative stress, impacting central metabolic pathways and membrane-associated functions regardless of oxygen availability.

## 1. Introduction

Tellurium is a group VI metalloid rarely found in its elemental form (Te^0^), and exists in nature mostly as soluble oxyanions, such as tellurite (TeO_3_^2−^), which is highly toxic to multiple organisms, especially prokaryotes. Although tellurium is present only in trace amounts in living organisms and shares chemical properties with selenium, sulfur, and oxygen, whether it plays a biological role remains unknown [1,2]. Nevertheless, its ability to inhibit bacterial growth is well-documented [3]. Compared with other oxyanions or metals, tellurite salts (e.g., K_2_TeO_3_) inhibit the growth of *Escherichia coli* at concentrations as low as ~1 µg/mL, which is ~100 times lower than the inhibitory concentration of other toxic metals and metalloids [4].

Multiple genes conferring resistance to tellurite have been identified in bacteria, located either on plasmids or chromosomes, although their mechanisms of action remain partially understood. For example, IncHI2 plasmids such as pMER610 carry the *ter* operon, which confers resistance not only to high tellurite concentrations but also to certain bacteriophages and colicins [5,6,7,8]. Another example is the *kilA* operon (including *klaA*, *klaB*, and *telB*), which is involved in plasmid maintenance and has been associated with tellurite resistance [9]. Additionally, the chromosomal *ars* operon—originally from plasmid R773—contributes to tellurite reduction via the ArsC reductase, which also reduces related oxyanions. Endogenous *E. coli* genes such as *tehA*/*tehB* [10] and membrane transporters like PitA (phosphate transporter) and ActP (acetate permease) have also been implicated in tellurite uptake and resistance [11,12,13].

Under aerobic conditions, tellurite toxicity has been closely associated with oxidative stress, mainly through the reduction in cellular thiols—such as glutathione and cysteine—which leads to the production of reactive oxygen species (ROS), including superoxide and hydrogen peroxide [1,3]. In response to ROS accumulation, *E. coli* induces protective mechanisms to limit damage to macromolecules. For instance, tellurite-exposed cells exhibit increased expression of antioxidant enzymes [14], and mutations in oxidative stress regulators affect tellurite sensitivity [15]. Tellurite also inactivates enzymes by oxidizing their cofactors; for example, it causes in vivo oxidation of protein thiols and disrupts [4Fe–4S] clusters in enzymes such as aconitase and fumarase, leading to metabolic blockage [15]. Moreover, tellurite interferes with heme biosynthesis, causing the accumulation of the toxic intermediate protoporphyrin IX [16]. Transcriptomic and metabolomic studies have further revealed that tellurite exposure disrupts multiple metabolic pathways—including glycolysis, the tricarboxylic acid cycle (TCA), cellular respiration, and glutathione metabolism—highlighting the multifaceted impact of this metalloid [2,15,16].

On the other hand, tellurite toxicity is markedly lower under anaerobic conditions. Early observations by Tantaleán et al. (2003) [17] reported that the MIC (minimum inhibitory concentration) of tellurite for *E. coli* grown anaerobically in rich medium was approximately 10-fold higher than in aerobic cultures. In our own assays using minimal medium, *E. coli* tolerated up to 100-fold higher concentrations under anaerobiosis compared to aerobic conditions. This pronounced difference indicates that, in the absence of oxygen, canonical ROS production is not expected, and thus, alternative mechanisms are likely responsible for the observed toxicity. Even under anaerobic conditions, *E. coli* is capable of taking up tellurite and chemically reducing it to elemental tellurium (Te^0^), which is typically observed as black precipitates in the medium. The efficiency of this reduction process has been shown to increase under anaerobiosis, possibly due to a higher availability of intracellular thiols and, potentially, specific reductive enzymatic activities. This enhanced reduction correlates with increased cellular tolerance. Therefore, in the absence of oxygen, *E. coli* appears to favor the conversion of tellurite into less reactive elemental deposits via thiol-dependent or enzymatic pathways, mitigating its cytotoxic effects [1,3,16]. These observations highlight the need to better understand the ROS-independent targets of tellurite toxicity. Specifically, which cellular pathways and molecular components are disrupted under anaerobic conditions, and to what extent these stress responses overlap with those activated during aerobic exposure?

To address this, we employed a comprehensive approach to characterize the impact of tellurite on *E. coli* in the presence and absence of oxygen. We hypothesized that tellurite exerts some fundamental toxic effects regardless of oxygen availability, and that *E. coli* activates similar stress response pathways under both aerobic and anaerobic conditions. We combined chemical-genomic screening (barcode-directed mutant fitness profiling), untargeted metabolomic analysis, and targeted biochemical assays to identify the damage caused by tellurite and the cellular response to this damage under aerobic vs. anaerobic conditions. By comparing these conditions, we aimed to pinpoint which effects of tellurite are ROS-independent (common to both aerobic and anaerobic stress) and thus intrinsic to tellurite’s chemistry. Our findings reveal that tellurite’s toxicity extends to critical metabolic processes—particularly amino acid, nucleotide, and lipid metabolism—and that *E. coli* adapts by remodeling its metabolism and membrane composition in the absence of oxygen. This work provides new insights into tellurium biology and bacterial stress physiology, highlighting mechanisms of toxicity that may be relevant to other redox-active compounds and to the development of strategies for tellurite resistance or remediation.

## 2. Results

### 2.1. Tellurite Resistance of E. coli Under Aerobic vs. Anaerobic Conditions

Consistent with previous reports [16,17], we found that *E. coli* is significantly more resistant to tellurite in the absence of oxygen. In our assays, the minimum inhibitory concentration (MIC) of tellurite for *E. coli* BW25113 was approximately 1 µg/mL (~4 µM), at aerobic growth, independently of the assayed medium—rich LB and minimal M9 media—in agreement with the previous reports [3]. In contrast, under anaerobic conditions, the apparent MIC depended on the medium. Using LB medium, the bacterial growth was inhibited by 10 µg/mL tellurite, 10-fold more metalloid compared to the aerobic MIC, whereas in minimal M9 medium, the MIC ranged from ~89 to 120 µg/mL, an increase of roughly (~100-fold. Thus, *E. coli* can tolerate higher tellurite concentrations in the absence of oxygen. The difference between rich and minimal media suggests that the components of LB (such as amino acids or peptides) may partially mask tellurite’s effect, perhaps by chemically interacting with tellurite or by helping cells cope [16,18]. To avoid this issue, we conducted most subsequent experiments in minimal media (M9 or MOPS), where tellurite’s impact is more direct and pronounced. All anaerobic experiments were conducted in minimal medium supplemented with a suitable electron acceptor (nitrate) to support anaerobic respiration, unless noted otherwise.

Rather than relying solely on MIC, which is determined in a very early growth phase, we also examined the effect of tellurite on established exponential-phase cultures by determining the minimum bactericidal concentration (MBC_50_), as described in the Methods section. We found that the concentration of tellurite required to kill 50% of the cells at OD 0.5 was ~5 µg/mL under aerobiosis, compared to ~20 µg/mL under anaerobiosis (Figure 1A). This four-fold increase in MBC_50_ is consistent with anaerobic cultures enduring higher tellurite exposure.

To understand if the reduced uptake of tellurite might underlie the increased anaerobic tolerance, we quantified tellurium in cell pellets and supernatants over time (Figure 1B). In both aerobic and anaerobic cultures treated with high tellurite doses (aerobic: 150 µg/mL, anaerobic: 200 µg/mL), the vast majority of tellurium remained in the supernatant fraction as tellurite or soluble Te species. However, at each time measured (1, 2, and 3 h), the percentage of tellurium that entered the cells was slightly lower under anaerobic conditions. For example, after 2 h under aerobic conditions, cells had taken up ~8.1% of the total tellurium, whereas in an anaerobic environment, the cells had incorporated only ~6.2%. By 3 h, these values were ~9.8% (aerobic) and ~8.0% (anaerobic). Correspondingly, a dark precipitate consistent with elemental tellurium (Te^0^) was visually observed in the culture medium. Although ICP analysis does not distinguish between oxidation states, the low-soluble tellurium content in the supernatant, together with the visual evidence, suggests extracellular reduction to elemental Te^0^, particularly under anaerobic conditions. Overall, we estimated ~12% less tellurite accumulation inside cells under anaerobic conditions. This modest difference in uptake suggests that while anaerobic cells may import or process tellurite slightly less, the primary reason for their higher tolerance lies elsewhere (i.e., how the cell handles the toxicant after uptake).

We next measured the effect of tellurite on intracellular thiols (Figure 1C). Untreated control cultures behaved as expected: aerobic cells gradually showed a decline in reduced thiols over time, likely due to the typical metabolic production of reactive oxygen species (ROS) [19], whereas anaerobic cells maintained steady thiol levels in the absence of oxidative stress. Upon exposure to tellurite, both aerobic and anaerobic cultures experienced a significant decrease in thiol content compared to their controls. By one hour of treatment, the total reduced thiols in aerobic cells dropped sharply (to ~50% of the initial value), and those in anaerobic cells also decreased (to ~70% of the initial value). After 2 h, thiols remained low in both aerobic and anaerobic samples, and continued to be depressed in the latter. These data are consistent with the hypothesis that tellurite chemically interacts with and oxidizes cellular thiols, even in the absence of oxygen. Notably, anaerobic cells started with a higher baseline thiol level (likely due to the lack of prior oxidative stress) and retained more reduced thiols after tellurite treatment compared to aerobic cells. This aligns with the idea that abundant thiols under anaerobiosis can buffer tellurite via chemical reduction (Painter reaction: RSH + TeO_3_^2−^ → RS–Te–SR → RSSR + Te^0^). In summary, tellurite exposure depletes cellular thiols in *E. coli* under both conditions; however, anaerobic cells, having a more reducing intracellular environment, may use excess thiols—possibly both chemically and enzymatically—to convert tellurite into inert Te^0^, which could contribute to reducing its toxic effects.

We also found that tellurite perturbs the cellular NADH/NAD^+^ ratio (Figure 1D). In the absence of tellurite, the NADH/NAD^+^ ratio was similar in aerobic and anaerobic exponentially growing cells (around 0.5–0.6, reflecting a balance between catabolic and anabolic demands). However, with tellurite treatment, a striking difference emerged: aerobic cells showed a significantly lower NADH/NAD^+^ ratio (~3.6-fold lower than the untreated cells), whereas anaerobic cells maintained a higher ratio (only slightly reduced from the untreated cells). This suggests that under aerobic conditions, NADH is rapidly consumed during the reduction in tellurite, possibly via flavoprotein oxidoreductases such as NDH-2 and nitrate reductases, generating ROS and exacerbating redox imbalance. Additionally, ROS generated from NADH oxidation may contribute to further NAD^+^ accumulation and oxidative damage. In contrast, anaerobic cells, despite facing metabolic stress, maintained a higher NADH pool, likely due to a combination of fermentative metabolism and lower ROS production.

Together, these experiments confirm that anaerobiosis drastically increases *E. coli*’s tolerance to tellurite. Anaerobic cells take up slightly less tellurite and more readily reduce it to elemental form (black precipitate) than aerobic cells, consistent with enhanced thiol-based tellurite reduction and lower intracellular thiol depletion in the absence of oxygen, as evidenced by increased elemental Te accumulation and higher residual thiol levels under anaerobic conditions (see Figure 1B,C). Nevertheless, even anaerobic cells suffer from thiol loss and growth inhibition upon tellurite exposure, indicating that ROS-independent damage also occurs. In the following sections, we investigate the nature of these additional stress mechanisms.

### 2.2. Evidence for Tellurite Targeting [4Fe–4S] Cluster-Containing Enzymes

One proposed mechanism of tellurite toxicity under aerobic conditions is the inactivation of enzymes containing [4Fe–4S] clusters, primarily through ROS-mediated oxidative damage [15]. Enzymes such as FumA and AcnB are known to be highly sensitive to such stress. However, previous studies have shown that in the absence of oxygen, no significant inactivation of these enzymes was observed. Considering this, we designed experiments to assess whether tellurite still affects [4Fe–4S] enzymes under anaerobic conditions through alternative, oxygen-independent pathways.

First, we compared *E. coli* growth on two different carbon sources in the presence of tellurite: glucose vs. succinate. Growing on glucose requires a fully active Krebs cycle (glycolysis produces pyruvate, which must be processed by pyruvate dehydrogenase, and the cycle enzymes aconitase, fumarase, etc., are needed for energy production). In contrast, growing on succinate feeds directly into the TCA at the level of succinate dehydrogenase, thereby bypassing the need for upstream cluster-containing enzymes, such as aconitase. If tellurite specifically impairs those cluster-dependent steps, cells growing on glucose should be more sensitive than those on succinate under aerobic conditions. We grew *E. coli* in MOPS medium with either glucose or succinate, under aerobic and anaerobic conditions, adding increasing concentrations of tellurite and measuring the area under the growth curve (AUC) (Figure 2A,B). Indeed, under aerobiosis, we observed that succinate-grown cultures were more resilient: for example, at 5 µg/mL tellurite, glucose cultures retained only ~25% of their growth (AUC), whereas succinate cultures retained ~64%. As tellurite increased to 10 µg/mL, succinate cultures still had ~31% AUC vs. only ~15% in glucose cultures. This result indicates that when bacteria utilize succinate (thus avoiding heavy reliance on aconitase/fumarase), they perform much better under tellurite stress in the presence of oxygen. Under anaerobic conditions, the trend was similar: both glucose and succinate cultures showed reduced growth with higher tellurite, but glucose-fed cells were slightly more inhibited (e.g., at 5 µg/mL tellurite, glucose cultures ~51% AUC vs. succinate ~48%; at 10 µg/mL, glucose ~29% vs. succinate ~19%). The differences anaerobically were smaller, perhaps because in the absence of oxygen, succinate utilization itself is limited by the anaerobic electron transport (since succinate dehydrogenase is part of the aerobic respiratory chain). Nonetheless, the aerobic data strongly support that tellurite disrupts enzymes of central carbon metabolism, consistent with ROS-mediated damage to [Fe–S] clusters under oxidative conditions.

Next, we examined specific mutants in genes involved in [Fe–S] cluster assembly and repair. *E. coli* has two main systems: the ISC (iron–sulfur cluster) system and the SUF (sulfur assimilation) system, along with accessory factors like HscB/HscA chaperones [20]. We reasoned that if tellurite damages Fe–S clusters, mutants defective in cluster assembly might show altered sensitivity. We tested multiple deleterious mutants related to the [Fe–S] cluster mechanism—Δ*iscA*, Δ*sufC*, Δ*sufD*, Δ*sufE*, and Δ*hscB*—assayed in the presence of tellurite (Figure 2C,D). Unexpectedly, under aerobic conditions, several of these mutants were more resistant to tellurite than the wild type. For instance, at 5 µg/mL tellurite, wild-type AUC was ~27% of control, whereas Δ*iscA* had ~107%, Δ*sufC* ~141%, Δ*sufE* ~132%, Δ*hscB* ~81%, and Δ*sufD* ~51% (the latter two showing partial resistance). Quantitatively, under aerobic conditions with 5 µg/mL tellurite, wild-type cells retained ~27% of control growth, while ΔiscA mutants retained ~107%, ΔsufC ~141%, and ΔsufE ~132% of control growth. These findings highlight the strong resistance phenotypes of these mutants. These mutants exhibited significantly better growth at various tellurite doses compared to the wild type. One interpretation is that in the absence of specific cluster assembly proteins, cells may possess fewer active Fe–S enzymes susceptible to inactivation by tellurite, thereby partially limiting its toxic effects. For example, deletion of the *iscA* gene could trigger compensatory mechanisms in iron handling or antioxidant response, potentially mitigating ROS generation. Interestingly, Δ*sufC* and Δ*sufD* mutants were notably resistant under aerobic conditions, suggesting that disruption of the SUF pathway might alter the cellular response to tellurite. These observations support further investigation into whether tellurite interferes with Fe–S cluster assembly dynamics or promotes maladaptive repair cycles, which will be addressed in the discussion.

Under anaerobic conditions, the effects of these mutations were less pronounced. Most mutants exhibited similar behavior to the wild-type strain, with only slight differences. One standout was Δ*iscA*, which showed higher tolerance than WT in anaerobic tellurite exposure (for instance, ~38% higher AUC at 5 µg/mL). Δ*sufC* under anaerobiosis was about as sensitive as WT, unlike its strong resistance under aerobiosis. This suggests that under anaerobic conditions, the presence or absence of these assembly factors is less critical, possibly because cluster damage is attenuated when ROS are minimal.

### 2.3. Chemical-Genomic Profile of Tellurite Toxicity: Common Genes Under Aerobiosis and Anaerobiosis

To identify the genes involved in tellurite resistance, irrespective of oxygen, we carried out a pooled mutant fitness screen (chemical-genomic profiling) under both aerobic and anaerobic conditions. After sequencing barcode abundances and computing CG z-scores for each mutant, we focused on the “common genes” that were significantly enriched or depleted in both conditions, meaning knockout gene mutants where tellurite tolerance is affected with or without oxygen (Figure 3A,B). Remarkably, a substantial number of mutants showed consistent effects in both environments: 86 mutants were negatively enriched (their loss made cells more sensitive to tellurite in both conditions), and 81 mutants were positively enriched (their loss made cells more resistant) in both conditions (Figure 4A,B). This overlap highlights a core tellurite response that is independent of oxygen presence.

Gene ontology enrichment analysis of the common negatively enriched genes (i.e., those required for full tellurite tolerance) revealed several functional categories (Figure 4C). One prominent category was bacterial-type flagellum assembly (GO:0044780). Multiple flagellar genes (e.g., *fliK*, *fliH*, *fliR*, *flgJ*, encoding structural or regulatory components of the flagellum) appeared in the sensitive set. This suggests that mutants lacking flagellar components were disproportionately sensitive to tellurite, implying that an intact flagellar apparatus or motility may aid in coping with tellurite stress. Flagella could help by dispersing cells or by some signaling crosstalk; interestingly, flagellar gene expression is energy-intensive, and its connection to tellurite tolerance merits further study. Another enriched category was the peptide metabolic process (GO:0006518) and the amino acid metabolic process (GO:0046416). Indeed, we identified several genes involved in amino acid utilization and transport in the sensitive set, including *dsdA* (D-serine ammonia-lyase), *tdcB* (threonine dehydratase), and *cysK* (cysteine synthase), as well as others related to amino acid catabolism, which were required for tolerance. This suggests that amino acid metabolism is a key factor in surviving tellurite. The ability to metabolize certain amino acids or to reconfigure metabolism helps detoxify reactive intermediates or generate alternative energy when tellurite disrupts primary pathways. In terms of broad functional groups (UniProt keywords), the negatively enriched genes were overrepresented in those encoding methyltransferases and S-adenosyl-L-methionine (SAM)-dependent enzymes, as well as components of the Krebs cycle (TCA). The enhancement of SAM-dependent enzymes is intriguing, as it could suggest that methylation processes (potentially related to lipid metabolism or tRNA modifications) serve a protective role during tellurite stress. The TCA connection aligns with our earlier observation that TCA activity (aconitase/fumarase) is a target of tellurite; mutants in some TCA genes (like *acnB* encoding aconitase or *sucA*/*B* encoding 2-oxoglutarate dehydrogenase components) showed increased sensitivity, reinforcing that tellurite challenges central metabolism.

For the common positively enriched genes (those whose deletion improved growth under tellurite in both conditions), enriched GO terms included components of the cytoplasm (GO:0005737). However, more revealing was the UniProt keyword enrichments, which highlighted amino acid transporters and proteases/metalloproteases. In fact, many of the top positively enriched hits were genes encoding amino acid transport or uptake systems. For example, deletions of *metI* (methionine transporter subunit), *pheP* (phenylalanine permease), *artJ* (arginine transporter periplasmic binding protein), *yehX* (glycine betaine uptake ATPase), *brnQ* (branched-chain amino acid transporter), *cydDC* (cysteine exporter), and others all conferred a tellurite resistance advantage. This suggests that limiting the influx or efflux of certain amino acids can be beneficial under tellurite stress. One explanation is that the presence of certain amino acids exacerbates toxicity—for instance, if tellurite causes the accumulation of toxic amino acid derivatives, reduced import might help. Alternatively, cells lacking these transporters may activate compensatory stress pathways that incidentally protect against tellurite. Similarly, multiple protease genes were in the positively enriched set: notably, *clpP* (ATP-dependent Clp protease proteolytic subunit), *pepP* (Xaa-Pro aminopeptidase), *mepM* (a murein endopeptidase), and *ptr* (protease III) were hits. The absence of these proteases made cells more tolerant to tellurite, which suggests that uncontrolled proteolysis during stress may be detrimental. Proteases could degrade key proteins or regulators that help cells adapt to stress; for instance, ClpP usually helps remove misfolded proteins; but without it, cells might build up chaperones or other protective proteins. Another angle is that the deletion of certain proteases induces the heat shock or SOS response, preconditioning cells to handle stress [21]. In any case, the involvement of proteases and amino acid transporters indicates that protein turnover and amino acid homeostasis are central to the tellurite response.

We compiled a list of the common genes affected by tellurite (Appendix A). A few notable examples from the negatively enriched (sensitive) list include *ackA* (acetate kinase, important in acetate metabolism), *narZ* (nitrate reductase Z, an anaerobic respiratory enzyme), *hybO* (hydrogenase-2 small subunit), *tonB* (a membrane energy transducer for nutrient transport), *rarA* (replication-associated recombination protein), and *radA* (DNA repair recombinase). The presence of *narZ* and *hybO* suggests that anaerobic respiratory functions are indeed affected by tellurite, which aligns with tellurite possibly interfering with anaerobic electron transport chains or the enzymes themselves. DNA repair proteins (RarA, RadA), which are required for tolerance, suggest that tellurite inflicts DNA damage, potentially via indirect routes such as metabolic imbalances or, under aerobic conditions, through endogenous ROS generation and Fenton-type reactions [15]. From the positively enriched (resistant when deleted) list, beyond the transporters and proteases mentioned, we also observed *ivbL* (a leader peptide for the isoleucine–valine biosynthesis operon), which suggests that downregulating branched-chain amino acid biosynthesis might be beneficial under tellurite (perhaps because it conserves NADPH or prevents the accumulation of certain intermediates). Additionally, *hcaB* (a dehydrogenase involved in 3-phenylpropionate degradation) was identified as a hit, suggesting that diverting metabolism away from nonessential pathways could be beneficial.

In summary, the chemical-genomic screening revealed multiple functional categories that influence *E. coli*’s response to tellurite under both aerobic and anaerobic conditions. These include genes involved in motility and flagellar biosynthesis, amino acid and peptide metabolism, and enzymes participating in central metabolic pathways such as the TCA. Additionally, the deletion of certain transporters and proteases correlated with increased tolerance, suggesting their potential role in modulating cellular stress. These findings point to multifactorial toxicity that extends beyond ROS-mediated effects and involves broader physiological processes.

### 2.4. Metabolomic Changes Induced by Tellurite in Aerobic and Anaerobic Conditions

To complement the genomic data, we analyzed the intracellular metabolites of tellurite-stressed cells. We were particularly interested in identifying metabolites that change in abundance due to tellurite, regardless of oxygen availability, as these could indicate ROS-independent effects on metabolic pathways.

Our untargeted metabolomic profiling identified dozens of metabolites significantly altered by tellurite. Table 1 (common metabolites) summarizes those metabolites whose levels changed in the same direction under both aerobic and anaerobic treatment. One clear outcome was that tellurite exposure perturbed amino acid metabolism. For instance, 5-phosphoribosylamine, an early intermediate in purine nucleotide biosynthesis, increased in abundance in tellurite-treated cells under both conditions (log_2_ fold-change ≈ +2.66 in aerobic, +1.07 in anaerobic). This metabolite is part of the purine biosynthetic pathway, suggesting that tellurite might slow the conversion of 5-phosphoribosylamine downstream (perhaps by affecting a subsequent enzyme or causing feedback that accumulates this intermediate). The accumulation of 5-phosphoribosylamine regardless of oxygen hints at a block in nucleotide biosynthesis, potentially caused by tellurite.

Another metabolite, iminoaspartate (also known as iminosuccinate, an intermediate in NAD^+^ biosynthesis from aspartate), showed a decrease under aerobic conditions (log_2_ FC ≈ −1.27) but a slight increase under anaerobic conditions (≈+1.11). This oxygen-dependent difference suggests tellurite may differentially impact NAD^+^ biosynthesis or aspartate utilization depending on oxygen, perhaps because aerobic conditions generate ROS that inactivate some NAD^+^ synthetic enzymes [22]. Interestingly, iminoaspartate was identified as significantly changed in both conditions (one down, one up), indicating a disruption in NAD metabolism in either direction.

A crucial finding was the change in membrane lipid-related metabolites. L-1-phosphatidyl-ethanolamine (L-PE), a key phospholipid (specifically the immediate precursor in the pathway that converts phosphatidylserine to PE or in cardiolipin biosynthesis), was strongly decreased in aerobic tellurite-treated cells (log_2_ FC ≈ −2.61), while it was moderately increased in anaerobic tellurite-treated ones (≈+1.68). Similarly, L-arginine levels dropped significantly under both conditions (−2.16 in aerobic, −1.12 in anaerobic, both *p* < 0.01), indicating arginine depletion. Arginine could be consumed in stress responses (polyamine synthesis like putrescine via arginine decarboxylase, might increase under stress). Its catabolic intermediate, N-succinyl-L-glutamate semialdehyde (in the arginine succinyltransferase pathway), also decreased in both cases (suggesting arginine degradation was hampered). Together, this implies that tellurite might be inhibiting parts of arginine biosynthesis or degradation, leading to arginine shortfall.

We also detected changes in specific phospholipid species. For example, a particular phosphatidylglycerol (PG) species with fatty acyl composition i-13:0/i-21:0 was decreased under both conditions (log_2_ FC ≈ −1.92 aerobic, −2.81 anaerobic), as was a phosphatidylserine (PS) species (22:2/22:6) which increased dramatically in aerobic (log_2_ FC +7.44) but decreased in anaerobic (−3.28) (Figure 5). The divergent behavior of that PS suggests that under aerobic stress, perhaps PS decarboxylation to PE is impaired (causing PS to accumulate), whereas anaerobically that step might proceed or PS is used differently. The significant buildup of polyunsaturated phosphatidylserine in aerobic cells exposed to tellurite is remarkable and may suggest an interference in the enzyme Phosphatidylserine decarboxylase, which relies on PLP (Pyridoxal 5′-phosphate). This situation may relate to the availability or functionality of PLP being influenced by tellurite, as discussed further below.

Table 2 (condition-specific metabolites) lists metabolites that changed significantly in only one condition. The majority of these were related to lipid metabolism. Under aerobic conditions, unique negative changes were observed in several PG and phosphatidic acid (PA) species with diverse fatty acid compositions. For instance, PG(16:0/20:3) and PG(i-13:0/i-19:0) were significantly diminished only in aerobic cells. This suggests that tellurite causes a remodeling of the membrane lipid profile when ROS are present, likely reflecting oxidative damage to specific fatty acyl chains or altered membrane enzyme activities. We also saw L-tryptophan significantly decreased only under aerobic conditions (log_2_ FC −1.16, not changed anaerobically). This might be due to oxidation of tryptophan or increased use of tryptophan in producing stress signals (or protein synthesis to replace damaged proteins, which would be more active in aerobic stress). S-lactoylglutathione, a product of methylglyoxal detoxification (via Glyoxalase I), decreased under aerobic conditions (−1.29 log_2_ FC), hinting at possible methylglyoxal stress interplay under oxidative conditions.

Under anaerobic conditions, some metabolites were uniquely affected. For example, 2′-deoxycytidine-5′-monophosphate (dCMP) was significantly reduced only under anaerobic conditions (log_2_ FC −1.86). This may suggest that tellurite disrupts DNA precursor pools, particularly under anaerobic conditions. In aerobic cells, any DNA damage initiates salvage pathways that could replenish dCMP. In anaerobic cells, although the active repair of DNA damage may not occur, there could still be a reduction in dCMP levels, possibly resulting from the repair of spontaneous damage or imbalanced synthesis. An additional distinct anaerobic response was observed in specific phospholipid derivates: for instance, PA (20:5/24:1) showed a significant decrease (−3.19 log_2_ FC), along with several other PA and PG species; meanwhile, a marked increase in PA(20:4/24:1) (log_2_ FC +1.68) and a phosphatidylethanolamine (PE) (18:4/P-18:1) (log_2_ FC +2.16). These findings suggest that alterations in membrane composition may contribute to the adaptation of anaerobic cells to tellurite stress. The overall pattern observed was that tellurite led to a reduction in several common phospholipids (such as PA and PG) while resulting in an increase in atypical or modified lipids under both conditions.

A particularly striking observation in the aerobic metabolome was the accumulation of methylated phosphatidyl-ethanolamine derivatives. We detected N-methyl-phosphatidylethanolamine and N,N-dimethyl-phosphatidylethanolamine (denoted as PE-NMe and PE-NMe_2_) that were highly enriched in tellurite-treated aerobic cells (e.g., PE-NMe_2_ (14:0/16:1) increased ~7.34 log_2_ FC; PE-NMe_2_ (16:1/16:1) +6.03 log_2_ FC; PE-NMe(16:1/18:0) +2.77). These are intermediates in the conversion of PE to phosphatidylcholine (PC) via trimethylation (a pathway common in some bacteria and eukaryotes using the enzyme phosphatidyl-ethanolamine N-methyltransferase, which uses SAM as a methyl donor) [23]. *E. coli* typically does not produce PC unless it has an exogenous pathway, but low levels of methylated PEs can occur. The dramatic increase here suggests that tellurite may be causing an accumulation of SAM or activating a usually minor methylation reaction. Alternatively, it is also possible that tellurite could interfere with the last metabolic step, (i) causing the accumulation of dimethyl-PE due to its inability to obtain a third methyl group, or (ii) it may have inhibited the conversion of PS to PE, indirectly resulting in these accumulations. In any case, the buildup of PE-NMe and PE-NMe_2_ implies an alteration in membrane lipid headgroup composition. Methylated PEs are more zwitterionic (approaching a PC-like structure), and their presence would increase membrane fluidity because they disrupt the tight packing of PE (PE tends to promote negative curvature and less fluidity, whereas PC is more bilayer-stabilizing but can increase fluidity due to its larger headgroup) [24]. The net effect observed—increased membrane fluidity—is consistent with these compositional changes. We surmise that tellurite stress prompts *E. coli* (especially under aerobic conditions) to attempt to modify its membrane lipids, either as a stress response or as a consequence of metabolic dysregulation (e.g., SAM pool imbalances or inhibition of certain lipases/transacylases causing unusual lipids to accumulate).

Taken together, the metabolomic data reinforce the idea that tellurite imposes a severe metabolic imbalance on cells. Key amino acids, such as arginine and tryptophan, are depleted, nucleotide biosynthesis is perturbed (with the accumulation of 5-phosphoribosylamine and a drop in dCMP), and lipid metabolism is drastically altered, resulting in a shift toward more fluid membrane constituents. Importantly, many of these changes occur under both aerobic and anaerobic conditions, indicating they are not simply due to ROS but to tellurite’s interaction with metabolic enzymes or cofactors. For example, the disruption of arginine and nucleotide synthesis might reflect tellurite targeting enzymes containing metal centers or cofactors (some enzymes in these pathways use iron–sulfur clusters or require reduced cofactors (NAD^+^/NADP^+^), which could be in short supply). The lipid changes suggest that tellurite interferes with enzymes like phosphatidylserine decarboxylase (which is PLP-dependent) or phospholipid transacylase, leading to the accumulation of precursors and unusual lipids. It is noteworthy that several of the processes highlighted here correspond to genes found in the CG screen:, e.g., metE (methionine synthase, a SAM-dependent methyltransferase) was a sensitive gene, and metabolomics indeed showed SAM-related reactions were affected; *cysK* (cysteine synthase) was a sensitive gene, and sulfur assimilation (APS levels) changed; transporters for amino acids (Met, Phe, Arg) were hits, and those amino acids showed changes.

### 2.5. Effect of Supplementing Key Metabolites on Tellurite Toxicity

If tellurite causes specific nutrient limitations (like arginine or tryptophan scarcity), providing those nutrients might rescue growth. Conversely, if the accumulation of certain metabolites is toxic (like methionine buildup), adding more might worsen outcomes. To test these hypotheses, we performed supplementation experiments.

Nucleotide precursors: We supplemented cultures with ATP, ADP, AMP, GDP, or UTP (each at 0.1 mM) and then exposed them to tellurite (Figure 6). Most nucleotide supplements did not markedly improve growth under tellurite stress, except for UTP under aerobic conditions. Aerobic cultures with a low concentration of tellurite (0.5 µg/mL) showed that adding UTP led to a higher %AUC than other nucleotide supplements or even the no-supplement control (which is a curious observation: UTP-supplemented + tellurite cultures sometimes grew better even than controls with no tellurite). This suggests that one limiting factor under tellurite stress might be UTP (or pyrimidine nucleotide) availability, perhaps due to tellurite inhibiting a step in pyrimidine biosynthesis. Extra UTP might bypass that bottleneck for RNA synthesis, thereby helping the cells maintain protein synthesis to counteract stress. However, this effect was only significant under aerobiosis; in anaerobic cultures, external UTP had no evident benefit (possibly because anaerobic cells were not as starved of UTP or because uptake and usage differ).

We also noticed from our CG data that genes related to pyridoxal phosphate (PLP) metabolism were negatively enriched. For example, genes such as *pdxB* or *yggS* (*serC*), which are involved in pyridoxal phosphate metabolism, showed negative enrichment in our dataset, although they are not explicitly listed in Appendix A. PLP (vitamin B_6_) is a known antioxidant against singlet oxygen and an essential cofactor in amino acid metabolism. We found that supplementing PLP significantly increased resistance to tellurite under aerobic conditions (growth up to 5 µg/mL tellurite was improved). Specifically, PLP supplementation raised the %AUC of cultures at 2–5 µg/mL tellurite by a noticeable margin compared to unsupplemented cultures. This can be explained by PLP’s ability to quench singlet oxygen, thereby mitigating ROS-mediated damage under aerobic conditions. In fact, vitamin B_6_ compounds have been shown to act as potent singlet oxygen quenchers and protect cells from photo-oxidative stress [25]. Our results suggest that under tellurite exposure (which generates ROS under aerobiosis), additional PLP provides an antioxidant boost, reducing oxidative damage. Moreover, PLP might stabilize or reactivate PLP-dependent enzymes that are inactivated by tellurite. This protective effect was absent under anaerobiosis (as expected, since there is little to no ROS to quench or scavenge, and PLP’s antioxidant role is not needed).

Amino acids: From metabolomic and CG insights, we focused our analysis on methionine, leucine, and phenylalanine. Importantly, we also tested amino acid supplementation in the absence of tellurite to evaluate baseline effects. Methionine supplementation alone showed minimal impact on growth under both aerobic and anaerobic conditions. Leucine supplementation alone caused a modest growth inhibition (~20–30% decrease in AUC aerobically and ~10–20% anaerobically), consistent with previous reports of branched-chain amino acid imbalance effects. Phenylalanine alone had negligible effects on growth. These observations confirm that the major inhibitory effects observed in combination with tellurite are synergistic rather than attributable solely to amino acid toxicity. When we added L-methionine to cultures, we observed a synergistic toxic effect with tellurite in both aerobic and anaerobic conditions. In aerobic cultures, even 0.5 mM Met combined with a very low tellurite dose (0.5 µg/mL) caused a severe drop in growth (%AUC ~14%, whereas tellurite alone was not so inhibitory). Similarly, in anaerobic cultures, Met plus 5 µg/mL tellurite resulted in significantly poorer growth compared to tellurite alone (Figure 7). This indicates that excess methionine exacerbates tellurite toxicity. One possible reason is that methionine can be oxidized to methionine sulfoxide by ROS [26], and perhaps tellurite facilitates that, consuming antioxidant resources (though under anaerobiosis, that logic does not apply due to lack of ROS). Another reason might be that methionine feeds into SAM (S-adenosylmethionine) production; too much SAM could drive aberrant methylation reactions or deplete ATP. Or, high methionine could trigger feedback inhibition in the cysteine biosynthesis pathway, lowering cysteine/GSH which are crucial for tellurite reduction. We measured intracellular Met levels: interestingly, tellurite-treated aerobic cultures showed a slight accumulation of Met (consistent with metabolomics and Table 3, which indicated Met went up). This accumulation might itself be harmful, and adding more Met externally exacerbates it.

Leucine supplementation also had a negative impact. Adding 0.5–1 mM leucine reduced the baseline growth (even without tellurite, high leucine can be inhibitory to *E. coli* growth by disrupting branched-chain amino acid homeostasis). In our tests, leucine addition caused a ~20–30% reduction in AUC aerobically and a ~10–20% reduction anaerobically. When combined with tellurite, leucine clearly enhanced toxicity under both conditions. For example, 0.5 mM leucine plus 0.5 µg/mL tellurite under aerobic conditions nearly abolished growth (whereas 0.5 µg/mL tellurite alone still allowed ~80% growth). And under anaerobic conditions, 5 µg/mL tellurite plus leucine caused a dramatic drop in AUC compared to tellurite alone (as detailed in Figure 7). Phenylalanine behaved similarly: it did not strongly affect growth by itself at moderate levels, but in combination with tellurite, it significantly worsened growth inhibition. Aerobic cultures with 0.5 µg/mL tellurite + Phe were severely inhibited (virtually no growth at 0.5 mM Phe, when commonly 0.5 µg/mL by itself causes maybe a 20% growth reduction). Anaerobically, 5 µg/mL tellurite + Phe caused a large drop in growth relative to tellurite alone. The intracellular levels of leucine and phenylalanine we measured aligned with these observations: in aerobic tellurite-treated cells, Leu and Phe pools were significantly decreased compared to untreated cells (consistent with metabolomics showing a drop in these amino acids under oxidative stress). This likely reflects increased usage or insufficient synthesis of these amino acids in response to tellurite stress. By adding Leu or Phe externally, one might expect to alleviate any shortage; however, our results showed the opposite effect on growth. This paradox could be due to regulatory effects—for instance, an excess of branched-chain amino acids might inhibit biosynthetic pathways (through feedback inhibition of enzymes or repression of operons like *ilv* genes), possibly leading to a shortage of other amino acids or accumulation of toxic intermediates [27]. Additionally, the catabolism of these amino acids may produce reactive oxygen species (ROS) or toxic ketoacids. An alternative viewpoint suggests that if tellurite induces a translational stall because of the depletion of specific amino acids, alleviating that stall could enable the cell to try protein synthesis, which may ultimately fail due to other forms of damage, leading to energy waste. Additionally, it is possible that high concentrations of leucine and phenylalanine, due to their hydrophobic nature, may exacerbate cellular stress by interfering with protein folding or membrane-associated processes under stressful conditions.

Methionine, leucine, and phenylalanine all appear to become toxic in the presence of tellurite, suggesting that tellurite-exposed cells exhibit impaired amino acid homeostasis. This is supported by our metabolomic data, which indicate that tellurite induces an intracellular imbalance—where certain amino acids accumulate to potentially toxic levels (e.g., Met), while others (e.g., Leu, Phe, Trp) may become limiting—potentially affecting protein synthesis capacity.

## 3. Discussion

Tellurite is an extremely toxic oxyanion for living organisms. Its toxicity in the presence of oxygen has been mainly associated with the generation of ROS and the concomitant oxidative damage that produces membrane lipoperoxidation, protein carbonylation, dismantling of iron–sulfur centers, and single- and double-strand breaks in DNA. However, ROS and the oxidation of macromolecules do not fully explain its toxicity within bacteria. Treatments with menadione (a superoxide-producing compound) or hydrogen peroxide are less toxic to bacteria than treatments with tellurite, suggesting that this oxyanion may exert some type of damage not associated with reactive oxygen species, which has been termed “direct tellurite damage.” Identifying targets unrelated to these species has proven challenging, as ROS quickly interact with any nearby macromolecules. The way to overcome this limitation is to work in anoxic environments, which normally require highly specialized equipment. Overcoming these limitations, important advances have been made in this field, identifying some metal(oid)s that produce ROS-independent toxic effects in *E. coli*, among which are the dismantling of solvent-exposed [4Fe–4S] centers by Hg(II), Ag(I), Cd(II), Zn(II), and Cu(I) [18]. Additionally, studies indicate that *E. coli* subjected to tellurite builds up protoporphyrin IX, which is harmful to the bacteria [16]. It is also noted that a transcriptional transition towards anaerobic metabolism is trigged [2], potentially due to the electron transport chain damage [28].

### 3.1. Tellurite Resistance of E. coli Under Aerobic vs. Anaerobic Conditions

Initially, tellurite-based MIC was evaluated in different culture media. Regardless of the medium used, the MIC of tellurite for *E. coli* under aerobiosis was the same, 1 μg/mL. However, under anaerobiosis, large differences were observed between the media used; the MIC was 10 and 100 times higher than under aerobiosis in LB and M9, respectively. We hypothesized that this difference is due to the specific batch composition of the LB medium, which contains an abundant and undetermined amount of amino acids and salts, which mask the effect of metals [16,18], while the minimal media M9 do not contain amino acids, and their salt composition is strictly defined.

The transcription factor Fnr is involved in the transition from aerobic to anaerobic metabolism, regulating the transcription of hundreds of genes. Since Fnr activates the transcription of the *pitA* gene, it was hypothesized that such differences in the MIC of tellurite and the viability of *E. coli* against this toxicant under both aeration conditions would be due to differences in the entry of tellurite into the bacteria. However, the results indicated that 1.8% more tellurite enters the bacteria under aerobiosis than under anaerobic conditions after 3 h of treatment (Figure 1B). Therefore, this discrepancy is not sufficient to assign the responsibility for its toxicity solely to the tellurite uptake.

Since no toxicity has been proven by the metalloid in its elemental state, the reduction in tellurite has been associated with its detoxification. This reduction, as mentioned above, is catalyzed by enzymes whose cofactor is NAD(P)H and by other cellular components that contain thiol groups. The latter, molecules containing thiols, are the first to be in contact with tellurite; therefore, evaluating the effect of tellurite on the intracellular concentration of these components provides a general view of the redox effect of the toxicant. Under aerobiosis, it has been described that the reduction in tellurite leads to a decrease in the cellular content of thiols, especially glutathione, which are essential for coping with oxidative stress [3,29]. When exposing *E. coli* cultures to tellurite, it was observed that, regardless of the aeration condition of the culture, the toxicant progressively oxidizes thiols and enzymatic cofactors, such as NADH (Figure 1C,D), causing the bacteria to lose their ability to cope with this stress.

The energy metabolism of *E. coli* is strongly compromised by tellurite. Estimating NADH/NAD^+^ levels in cultures treated with tellurite shows that the metalloid affects the oxidation state of these molecules and their production. Under aerobiosis, it has been described that, as a result of tellurite oxidative stress, there is an increased conversion of NADH to NAD^+^ [30], which explains the abrupt decrease in the NADH/NAD^+^ ratio (Figure 1D). On the other hand, the smaller decrease in this parameter under anaerobiosis could indicate that tellurite per se has an effect on the conversion of NADH to NAD^+^, independently of oxidative stress, either by directly oxidizing it as a cofactor for tellurite reductases and/or by affecting the expression and function of genes and proteins associated with its biosynthesis.

All the above-mentioned leads to the loss of functionality of enzymes that play important roles in bacteria and use these molecules as cofactors to carry out other cellular processes. Among these NAD(P)H-dependent enzymes, some have been described that possess tellurite reductase activity, such as the alkyl hydroperoxide reductase AhpF, among others [31].

### 3.2. Evidence for Tellurite Targeting [4Fe–4S] Cluster-Containing Enzymes

Previously, and also under aerobiosis, it was described that this oxyanion indirectly affects enzymes belonging to the electron transport chain and others that use [Fe–S] centers as a prosthetic group, including aconitases and fumarases belonging to the Krebs cycle and NADH dehydrogenase I of the electron transport chain [15,32]. In *Deinococcus radiodurans*, exposure to tellurite results in decreased synthesis of metabolic enzymes related to glycolysis and the Krebs cycle [33]. In this regard, our results are consistent, as the anaerobic GC analysis resulted in negative enrichment in several categories related to the bacteria’s energetic status, including anaerobic respiration and metabolic processes (Appendix A).

The negatively enriched genes related to the Krebs cycle were *fumA*, *prpC*, and *sucD* (Appendix A). FumA is a dimeric fumarase that uses a [4Fe–4S] center as a prosthetic group for its activity, which is affected by tellurite [15]. PrpC is a methylcitrate synthase that can also function as a citrate synthase. SucD is one of the two subunits of the succinyl-CoA synthetase enzyme involved in the Krebs cycle [34].

In summary, these results support the notion that tellurite’s toxicity involves interference with iron–sulfur cluster enzymes and/or their biogenesis. Bacteria growing in a metabolic mode that bypasses sensitive Fe–S enzymes (using succinate) were more resistant to tellurite, and mutations in Fe–S assembly machinery altered tellurite resistance (often conferring protection under aerobic conditions). This is consistent with earlier findings that tellurite can directly inactivate dehydratases and other Fe–S proteins [15,35] and with the idea that some cellular systems (like ISC/SUF) might inadvertently contribute to tellurite toxicity by attempting to repair clusters that then become re-damaged, consuming resources and releasing free iron. These resistance phenotypes likely reflect reduced Fe–S cluster biogenesis in the mutants, leaving fewer functional Fe–S enzymes for tellurite to damage. Tellurite is known to inactivate such enzymes, including aconitase, by oxidizing their [4Fe–4S] clusters. Paradoxically, this suggests that lacking Fe–S assembly components may reduce cellular vulnerability to tellurite. Moreover, the deletion of *iscA* or *suf* genes may trigger compensatory responses such as increased expression of antioxidant defenses or iron sequestration systems, potentially lowering ROS levels under stress. Notably, *iscA* deletion can activate the IscR regulator, inducing *suf* operon expression and enhancing stress resilience. While we did not directly measure Fe–S enzyme activities or ROS in these mutants, future studies could address these mechanisms.

### 3.3. Chemical-Genomic Profile of Tellurite Toxicity: Common Genes Under Aerobiosis and Anaerobiosis

Given the limited knowledge about tellurite toxicity under anaerobic conditions, a method with the lowest possible bias was used. This method, called genomic chemical profiling, consisted of a genome-wide deletion library of mutants; the selected gene was replaced with a kanamycin resistance cassette plus a unique barcode for each mutant [36]. The aim is to identify deletions that are detrimental to tellurite tolerance, thereby gaining insight into the mechanisms underlying its toxicity, as well as potential cellular response mechanisms.

Various bioinformatics analyses provided a comprehensive view of the interaction between tellurite and *E. coli* under both aerobic and anaerobic conditions. These analyses indicated that several metabolic pathways and cellular processes are involved in tolerance to this toxicant. Those cellular processes and metabolic pathways that were negatively enriched in these analyses represent cellular components that, in the wild-type *E. coli* strain, would contribute to tolerance to this toxin. If the opposite were the case, positive enrichments would most likely suggest that such deleted genes are cellular targets of tellurite in the wild-type strain. When analyzing the genes enriched under aerobiosis and anaerobiosis, it was found that the negatively enriched common terms correspond to methyltransferases and SAMs, the synthesis and assembly of the bacterial flagellum, amino acid and peptide metabolic processes, and the Krebs cycle (Appendix A).

Previous studies proposed tellurium methylation as a bacterial detoxification mechanism, mediated by methyltransferases. Methyltransferases capable of alkylating tellurium have been identified on the *E. coli* chromosome. One of these enzymes is present in the tellurite resistance determinant tehAB, which, when overexpressed from multicopy plasmids, confers tellurite resistance [10]. The proposed mechanism indicates that TehA plays a role as a transporter, while the TehB enzyme utilizes SAM for its tellurite methyltransferase activity [37]. The SAM-dependent thiopurine methyltransferase (TmpT) enzyme was also identified in *E. coli*, which had tellurium alkylating activity [38]. Methylated tellurium compounds, such as dimethyl tellurium (CH_3_-Te-CH_3_) and dimethyl ditellurium (CH_3_-Te-Te-CH_3_), are partly responsible for the characteristic “garlic odor” of bacterial cultures exposed to tellurite [39].

Of the identified methyltransferases, the MetE enzyme stands out, as it is involved in the last step of methionine synthesis. This enzyme is not SAM-dependent but uses L-homocysteine and 5-methyltetrahydropteroyltri-L-glutamate as substrates to generate L-methionine. *E. coli* subjected to oxidative stress in minimal medium develops auxotrophy for methionine because MetE is inactivated by the oxidation of the cysteines present in the enzyme [40].

The multifunctional enzyme CysG, responsible for siroheme synthesis, is not directly involved in tellurite resistance [41]. According to the results, the deletion of the gene for this enzyme and the concomitant lack of siroheme decrease *E. coli* tolerance to tellurite. Siroheme is one of the prosthetic groups found in sulfite and nitrite reductase enzymes, which generate sulfide and ammonium, respectively. Furthermore, sulfide assimilation is closely related to the synthesis of cysteine and glutathione, which contribute to tellurite resistance [42,43,44]. This suggests that when CysG is inhibited by tellurite, the bacteria lack sulfite reductase activity and, consequently, lose the ability to regenerate new thiol groups that can participate in reducing the toxin.

The other three identified methyltransferases are related to the ribosome and protein synthesis. The YfiC protein is responsible for the methylation of tRNA-Val. Mutants in the gene for this protein are sensitive to oxidative and osmotic stress [45]. RsmJ and RsmD are responsible for the methylation of 16S rRNA at positions G1516 and G966, respectively, affecting the interaction of tRNA within the ribosomal P site. Δ*rsmJ* mutants are cold-sensitive, while Δ*rsmD* mutants present an altered frequency of translation initiation without an associated phenotype [46,47].

The synthesis and assembly of the bacterial flagellum is another mechanism that could be impaired by tellurite exposure. It has been determined that microorganisms can adapt to stress conditions by losing their flagella and forming biofilms. In this regard, it has been determined that under aluminum stress, many flagellar genes (e.g., *flh*, *fli*, *flg*) in *Rhodanobacter* sp. are lost and/or down-expressed [48].

Another negatively enriched term is related to peptidases and amino acid degradation metabolism. Among the identified peptidases are dcp and pepT (Appendix A). Dcp is a periplasmic peptidase that hydrolyzes the peptide bond of small peptide substrates; its activity requires divalent ions such as Mn(II), Mg(II), Ca(II), and Co(II), while it is inhibited in the presence of Cu(II), Zn(II), and Ni(II) [49]. PepT is a peptidase that can degrade leucine, lysine, methionine, and phenylalanine when located at the amino terminus of certain tripeptides; its activity is dependent on an undetermined divalent cation, and its transcription increases under anaerobiosis when FNR binds to the respective promoter [50].

The *dsdA* gene codes for DsdA, an enzyme that catalyzes the deamination of D-serine, yielding pyruvate and ammonium [51]. DcyD is a D-cysteine desulfhydrase that protects the bacteria from growth inhibition by D-cysteine and also allows the use of this compound as a source of sulfur [52]. A common feature between these two enzymes is that they use pyridoxal 5-phospate, PLP, a common cofactor used by many metabolic enzymes.

In this regard, a study conducted in parallel in our laboratory linked the deletion of genes encoding chaperones and proteases with increased toxicity of soft metal(oid)s, suggesting that they induce protein aggregation in *E. coli.* The synthesis of chaperones and proteases would prevent the formation of protein aggregates, which, in the case of tellurite, primarily affects nascent proteins [53].

The broad array of cellular systems identified in our chemical-genomic analysis—including methyltransferases, amino acid degradation enzymes, sulfur metabolism factors, and chaperone pathways—reflect complex regulatory responses to tellurite stress. While our study did not include direct transcriptomic analyses, previous work has shown that tellurite exposure induces the expression of genes involved in oxidative stress defense, sulfur metabolism, and protein quality control. These findings align with the pathways uncovered in our metabolomic and mutant-based screens. We therefore recognize that transcriptomic approaches such as RNA-seq or qPCR would provide complementary insights by confirming gene expression changes underlying these phenotypic responses. However, considering the extensive coverage provided by our current system-level methods and the resource demands of full RNA-seq under both aerobic and anaerobic conditions, we focused this study on functional and metabolic outcomes. Future work may incorporate transcriptomic analysis to expand on the regulatory aspects suggested by our findings.

### 3.4. Metabolomic Changes Induced by Tellurite Under Aerobic and Anaerobic Conditions

It has been observed previously that the exposure of bacteria to a toxic compound can generate modifications in the plasma membrane. In *E. coli*, treatment with H_2_O_2_, Co^2+,^ or excess Cu^2+^ and Fe^2+^ in the presence of H_2_O_2_ has been determined to alter membrane composition, increasing the percentage of cardiolipin (CL) and decreasing the percentage of PG and PE; in some cases, even decreasing the total phospholipid content [54,55,56].

In early aerobic cultures of *R. capsulatus*, the membrane permeability integrity of ~50% of cells were compromised by subinhibitory amounts of tellurite. In late culture phases, this integrity was restored to 90%, which was correlated with an increase in its membrane potential [57]. This suggests that some effects of tellurite may be due to modifications in membrane structure (lipid/protein composition). In parallel, membrane collapse occurs due to the disappearance of the pH gradient and the consequent depletion of ATP [58]. Furthermore, the monounsaturated fatty acid (MUFA) content of *E. coli* growth under aerobic conditions and exposed to tellurite has been described as a substrate for lipid peroxidation, producing highly reactive aldehydes that induce protein carbonylation [59,60,61].

The change in the lipid and protein composition of the plasma membrane has been considered an adaptive mechanism against various types of stress, helping to tolerate acidic pH [62,63], cold in psychrophilic organisms [64], and increased temperature in mesophilic organisms [65]. Membrane fluidity depends on the length of the fatty acid present in the phospholipids—14 to 24 carbon atoms— and the level of unsaturation of the carbon chain; the shorter and more unsaturated the phospholipids, the more fluid the membrane. Long chains exhibit larger interactions, as the free energy between two chains decreases by 0.5 kcal/mol for each CH_2_ group added, thereby strengthening the overall interaction. On the other hand, membranes with saturated lipids are rigid, since the hydrocarbon chains interact strongly with each other. The presence of double bonds in the chain decreases the interaction, favoring membrane fluidity [66].

From the metabolomic analysis of *E. coli* exposed to tellurite in aerobic and anaerobic conditions, it can be inferred that a change occurs in the lipid composition of the plasma membrane. Several fatty acids that form part of the membrane of bacteria exposed to tellurite vary according to oxygen availability, including phosphatidic acid (PA), phosphatidylethanolamine (PE), phosphatidylglycerol (PG), phosphatidylglycerol phosphate (PGP), and phosphatidylserine (PS). During aerobic treatment with tellurite, the PG content decreases, while the content of PE and PS increases. In anaerobic culture, no defined molecular pattern was observed, but rather a mixture of fatty acids and phospholipids of different types, characterized by elongated chains and various degrees of unsaturation (Table 1 and Table 2, Figure 5).

### 3.5. Effect of Supplementing Key Metabolites on Tellurite Toxicity

As discussed in previous sections, it was mentioned that one of the factors that could be responsible for the difference in tellurite MIC between LB medium and M9-MOPS medium is their amino acid composition. The results of the analysis show positive enrichments in several metabolic pathways related to amino acid synthesis, particularly phenylalanine under aerobiosis and cysteine, methionine, arginine, leucine, isoleucine, and valine under anaerobiosis. In the presence of cysteine and tellurite, bacterial growth is drastically inhibited, a phenomenon similar to what Scala and Williams (1963) [67] reported when adding tellurite to culture media containing cysteine or homocysteine as a sulfur source (Appendix A). When methionine was supplemented, tellurite toxicity increased under both metabolic conditions, a phenomenon also previously observed by Scala and Williams (1962) [68]. On the other hand, aerobic cultures treated with tellurite tend to accumulate methionine in their cytoplasm (Table 3). When arginine and tellurite were supplemented to growing cultures, harmful effects on the culture were observed only in the presence of both compounds under aerobic conditions (Appendix A). However, metabolomic analysis indicates that arginine concentration decreases in tellurite-treated cultures, regardless of the presence of oxygen (Figure 5). In the case of tryptophan, its concentration decreased in aerobic cultures with tellurite, suggesting that this phenomenon could be related to ROS. Supplementing tryptophan to aerobic cultures with tellurite decreased bacterial fitness, suggesting some interaction with ROS.

Branched-chain amino acids (BCAAs) appear to be related to tellurite. Leucine had a toxic effect on the cultures, which was enhanced in the presence of tellurite. Under aerobiosis, cultures treated with tellurite tended to accumulate intracellular leucine (Table 3), while valine inhibited growth. When cultures were exposed to tellurite, the intracellular concentration of valine increased under aerobiosis (Table 3). Together, isoleucine and tellurite had a growth inhibitory effect enhanced by the presence of oxygen (Appendix A). Isoleucine levels decreased in aerobic tellurite-treated cultures (Appendix A).

The increase in tellurite toxicity in aerobic cultures with histidine or lysine (Appendix A) is due to the fact that these amino acids are prone to modification by aldehydes resulting from lipid peroxidation [69]. Lysine tends to accumulate in aerobic cultures treated with tellurite (Table 3).

To integrate the findings of our chemical-genomic and metabolomic analyses, we generated a schematic summary illustrating the proposed mechanisms of tellurite toxicity under both aerobic and anaerobic conditions (Figure 8). This model highlights key differences and overlaps in bacterial responses depending on oxygen availability. Under aerobic conditions, reactive oxygen species (ROS) contribute to tellurite toxicity through oxidative damage to lipids, proteins, and DNA. In contrast, under anaerobic conditions, oxidative stress is largely absent, yet toxicity persists due to metabolic imbalances, impaired amino acid biosynthesis, thiol depletion, and protein aggregation. Shared features across both conditions include the disruption of [Fe–S] cluster-containing enzymes, SAM-dependent methylation pathways, membrane lipid remodeling, and elemental tellurium formation. This graphical representation provides a concise conceptual framework for understanding the multifactorial nature of tellurite toxicity in *E. coli*.

In addition to the mechanistic insights discussed above, we compared our findings with known stress responses to hydrogen peroxide and heavy metals such as copper and mercury. Both tellurite and H_2_O_2_ can damage Fe–S cluster-containing enzymes and promote oxidative stress, but tellurite uniquely induces substantial remodeling of membrane lipid composition—particularly under anaerobic conditions—rather than just lipid peroxidation. Tellurite also triggers the accumulation of unusual phospholipid species and shifts in saturation patterns, which are not typical of H_2_O_2_ exposure. Regarding Fe–S cluster damage, tellurite indirectly inactivates [4Fe–4S] enzymes via oxidative thiol depletion and redox imbalance. Similarly, Cu^2+^ can generate ROS through Fenton-like reactions that damage Fe–S clusters, while Hg^2+^ preferentially binds cysteines, disrupting cluster assembly by displacing iron or sulfur ligands. However, tellurite’s dual role as an oxidant and a chalcogen compound with unique chemical reactivity results in a distinct stress signature. We observed perturbations in sulfur assimilation, nucleotide biosynthesis, and amino acid homeostasis, as well as heme precursor accumulation—responses not widely reported under Cu^2+^ or Hg^2+^ exposure. These differences likely reflect tellurite’s hybrid mechanism of toxicity. Conserved responses across these stressors include thiol depletion, Fe–S enzyme inactivation, and antioxidant defense induction, whereas the unique aspects of tellurite toxicity include membrane lipid remodeling and widespread metabolic imbalance. These insights are summarized in the schematic Figure 8.

## 4. Materials and Methods

### 4.1. Bacterial Strains and Culture Conditions

The strain used in this study (Table 1) was *Escherichia coli* K-12 BW25113 (National BioResource Project, Tokyo, Japan), the parent of the Keio knockout collection. For some experiments, isogenic knockout mutants from the Keio library were used (notably deletions in *sufB*, *sufC*, *sufD*, *iscA*, *iscU*, *hscB*, etc., as indicated in the Results). Cultures were routinely grown at 37 °C in either Luria–Bertani (LB)-rich medium or in defined minimal media as specified. Minimal media included M9 minimal medium [70] with 0.2% (*w*/*v*) glucose, or MOPS-defined medium (containing ammonium chloride, sulfate, calcium chloride, magnesium chloride, sodium chloride, micronutrients, and 40 mM phosphate buffer) supplemented with either 0.26% glucose or 0.4% succinate as carbon source. For anaerobic growth, cultures were handled inside an anaerobic chamber (Coy Laboratory Products Inc., Grass Lake, MI, USA) under a 100% N_2_ atmosphere. In anaerobic MOPS cultures, 20 mM potassium nitrate (KNO_3_) was added as an alternative terminal electron acceptor to support anaerobic respiration. When required for plasmid maintenance, kanamycin was added at 50 µg/mL. Potassium tellurite (K_2_TeO_3_, Sigma-Aldrich P0677 (St. Louis, MO, USA)) was prepared fresh in deionized water for each experiment and added at the indicated concentrations.

For consistency, overnight pre-cultures were grown aerobically in LB (or an appropriate medium) and then diluted into fresh medium for the experiments. Cultures were grown to mid-exponential phase (optical density at 600 nm, OD_600_ of ~0.5) before tellurite treatments, unless stated otherwise. Anaerobic cultures were pre-equilibrated in the chamber to ensure oxygen was eliminated.

### 4.2. Cell Viability

*E. coli* K-12 BW25113 (50 mL) was grown to OD_600_ ~0.5 under aerobiosis or anaerobiosis. A total of 950 µL of culture was taken and added to a 48-well plate, where 50 µL of tellurite had been previously added at a given concentration. At 0, 60, and 120 min, 20 µL was taken to make 1/10 serial dilutions in 96-well plates; then, 4 µL of each dilution was taken to make spots on LB agar plates. After 24 h of incubation at 37 °C, the colonies were counted. The CFU/mL was calculated using Equation (1).CFU/mL = (#colonies ∗ 10(^−dilution^))/0.004 mL(1)

### 4.3. Quantification of Tellurite E. coli Cultures

To quantify tellurite uptake and its reduction to elemental tellurium (Te^0^) over time, we treated cultures with a high sublethal concentration of tellurite and measured tellurium content in cell pellets vs. supernatants. *E. coli* BW25113 was grown in M9 to OD_600_ ~0.5 in either aerobic or anaerobic conditions. Tellurite was then added (150 µg/mL for aerobic cultures and 200 µg/mL for anaerobic cultures, chosen to yield similar partial growth inhibition in each condition), and samples were taken at 1, 2, and 3 h of exposure. Each 5 mL sample was centrifuged at 9000× *g* for 5 min at 4 °C to separate the cell pellet from the supernatant. The pellet was washed once with 1× PBS to remove extracellular tellurite. The supernatant and wash fraction (containing unabsorbed and extracellular tellurite or Te deposits) and the cell pellet were each digested in 2% HNO_3_ (65% nitric acid diluted to 2%) to dissolve tellurium. Tellurium content in each fraction was quantified by inductively coupled plasma optical emission spectroscopy (ICP-OES) using a PerkinElmer Optima 2000 DV instrument (PerkinElmer, Waltham, MA, USA), calibrated with Te standards from 0.01 to 250 µg/mL. The percentage of tellurium present in cells vs. supernatant was calculated for each time point. This allowed us to determine the fraction of tellurite that entered the cells and the extent of elemental Te precipitation (since any tellurite reduced to insoluble Te^0^ would appear in the pellet or as black particulate in the tube). All measurements were performed in triplicate, and values are reported as mean ± standard deviation.

### 4.4. Determination of Free Thiol Content

One known effect of tellurite is oxidation of reduced thiol groups (–SH) in molecules like glutathione (GSH) and cysteine. To evaluate this under both aerobic and anaerobic conditions, we quantified the total reduced thiols (RSH) in cell extracts before and after tellurite exposure using Ellman’s reagent (5,5′-dithiobis(2-nitrobenzoic acid), DTNB) [71]. Cultures of *E. coli* BW25113 were grown to OD_600_ ~0.5 (20 mL volume each). Each culture was split into two 10 mL aliquots: one aliquot would serve as an untreated control, and the other would be treated with tellurite. For the treated samples, we added 5 µg/mL tellurite for aerobic cultures and 20 µg/mL for anaerobic cultures (doses chosen to cause measurable stress without complete killing, roughly corresponding to the MBC_50_ under each condition). At 0 h (immediately before adding tellurite), 1 h, and 2 h after treatment, 1 mL of culture was withdrawn to quantify protein content, and 0.5 mL was withdrawn for thiol assay. These samples were centrifuged (14,000× *g*, 5 min), and the cell pellets were stored at −20 °C until processing. For the DTNB assay, pellets were resuspended in 1 mL of RSH assay buffer (50 mM Tris–HCl pH 8.0, 5 mM EDTA, 0.1% SDS) containing 0.1 mM DTNB. Suspensions were incubated at 37 °C for 30 min with occasional mixing, then centrifuged (14,000× *g*, 10 min). The absorbance of the supernatant was measured at 412 nm to detect the TNB product from DTNB reduction. Using a molar extinction coefficient of 13,600 M^−1^ cm^−1^ for TNB, the concentration of free thiols in each sample was calculated. Thiol concentrations were normalized to the total protein content of the sample (determined by a Bradford assay or BCA assay on the 1 mL sample taken in parallel). Results were expressed as µmol thiol per mg protein, and the percent thiol remaining in treated vs. control cells was calculated over time.

### 4.5. Determination of NADH and NAD^+^ Content

The cellular redox status (balance of NADH to NAD^+^) is a sensitive indicator of metabolic perturbation. To assess whether tellurite affects the intracellular redox state differently in aerobic vs. anaerobic conditions, we measured the NADH:NAD^+^ ratio in cultures with and without tellurite. Cultures (60 mL each) of E. coli BW25113 were grown to OD_600_ ~0.5 in M9 (aerobic or anaerobic). Each culture was divided into two 30 mL portions; one was immediately treated with tellurite (5 µg/mL for aerobic, 20 µg/mL for anaerobic), and the other received no tellurite (control). After 1 h of incubation, 1 mL was collected for protein quantification and 2 samples of 10 mL for the NADH/NAD^+^ assay. The samples were centrifuged at 9000× *g* for 10 min at 4 °C, suspended in 1 mL of 1× PBS, centrifuged again under the same parameters, and the cell pellet was frozen at −80 °C until processed. The determination of the NADH/NAD^+^ content was carried out via the “redox-cycling” assay described by Leonardo et al. (1996) [19] and modified by Kohanski et al. (2007) [30].

Cell pellets were suspended in 250 µL of 0.2 M HCl (NAD^+^ extraction) or 0.2 M NaOH (NADH extraction). The samples were heated to 100 °C for 10 min. They were then cooled and centrifuged at 14,000× *g* for 5 min at 4 °C. The supernatant was transferred to an amber Eppendorf tube to protect it from light and kept on ice. The calibration curve was prepared using serial dilutions between 1.5 mM and 0.02 mM of NADH and NAD^+^ (Sigma-Aldrich, St. Louis, MO, USA). A neutralizing buffer consisting of 0.1 M Bicine pH 8.0, 4 mM EDTA pH 8.0, 10% ethanol, and 25 mM HCl (NADH extraction) or 25 mM NaOH (NAD^+^ extraction) was used. To 50 µL of each sample, 110 µL of its corresponding neutralizing buffer was added. Subsequently, 4 µL of alcohol dehydrogenase (ADHII) 500 U/mL (Sigma) was added to each sample, followed by its incubation for 25 min in the dark at 30 °C. Then, a solution of phenazine ethosulfate (PES, Sigma) 1.66 mM and 3-(4,5-dimethylthiazol-2-yl)-2,5-diphenyltetrazolium bromide (MTT, Merk, Darmstadt, Germany) 420 µM were added. The reaction was immediately incubated at 30 °C for 10 min and monitored at 570 nm in the NanoQuant Infinite M200 Pro plate reader (Tecan, Männedorf, Switzerland). The reduction rate of MTT is proportional to the concentration of NADH or NAD^+^ in the sample. The results were expressed as the ratio between the concentration of NADH and NAD^+^, normalized by the protein concentration.

### 4.6. Growth Curve Analysis and Area Under Curve (AUC) Calculation

To examine growth dynamics under tellurite stress, we monitored the growth curves of *E. coli* and various mutant strains with and without tellurite and computed the percent area under the curve (%AUC) as a measure of overall growth fitness. Growth assays were conducted in 96-well microplates using a plate reader (37 °C with orbital shaking). For wild-type *E. coli* BW25113, overnight cultures were diluted into MOPS minimal medium with either 0.266% glucose or 0.4% succinate as the carbon source (to modulate metabolic pathways). These cultures were grown under either aerobic or anaerobic conditions and supplemented with a range of tellurite concentrations (0 to 20 µg/mL, as appropriate for the condition). OD_600_ was measured every 15 min for at least 12–24 h. For mutant fitness experiments, we similarly grew knockout strains (e.g., Δ*iscA*, Δ*sufC*, Δ*sufD*, Δ*sufE*, Δ*hscB*) in MOPS glucose medium with varying tellurite concentrations under aerobic or anaerobic conditions, recording growth kinetics.

The area under the growth curve (OD vs. time) was calculated for each condition using the Growthcurver R package (v0.3.0; [72]) or a custom script. The area was then normalized to the area of an untreated control curve (set as 100%) to yield %AUC. This metric captures both growth rate and yield. Each experiment was performed with 4–8 biological replicates. We used %AUC to compare how different conditions or mutants perform relative to the no-tellurite control.

### 4.7. Chemical-Genomic Profiling (Barcode Sequencing)

To identify genes involved in tellurite sensitivity or resistance, we performed a chemical-genomic (CG) screen using the *E. coli* Keio knockout collection tagged with unique barcodes [73]. A pooled library of ~3900 barcoded single-gene deletion strains was grown in batch culture under two conditions: aerobic + tellurite and anaerobic + tellurite. For each condition, we aimed to use a tellurite concentration that inhibits wild-type growth by ~20% (a sublethal stress allowing all mutants to grow, albeit at different rates). Based on preliminary tests, we chose 1 µg/mL tellurite for aerobic and 10 µg/mL for anaerobic conditions to achieve roughly 20% growth inhibition in bulk culture. Pooled mutant cultures were grown in triplicate in M9 minimal medium (with 0.2% glucose) under the respective conditions until the culture OD dropped relative to untreated control (indicating growth inhibition). Genomic DNA was then extracted from each pooled culture. The unique 20 bp barcode tags from each mutant (up- and down-tags) were PCR-amplified and subjected to Illumina sequencing to quantify each mutant’s abundance in the population (similar to the approach described by Typas et al., 2008 [74], and by subsequent barcode sequencing methodologies). Sequencing reads were mapped to the corresponding gene knockouts.

For each mutant, a CG z-score was calculated, representing the normalized change in abundance of that mutant in the tellurite-treated pool relative to an untreated control pool. The z-score normalization centers the data (mean 0, SD 1) so that most mutants cluster around zero (no effect). Mutants with CG z-score ≤ −1 were considered negatively enriched (meaning the mutant was under-represented after tellurite treatment, i.e., its absence caused a growth disadvantage, suggesting the gene is important for tolerance). Mutants with CG z-score ≥ 1 were considered positively enriched (the mutant over-represented, i.e., its absence conferred a growth advantage, suggesting the gene’s normal function contributes to tellurite toxicity or the gene is nonessential under stress). By focusing on genes that showed significant z-score shifts under both aerobic and anaerobic conditions, we could identify common factors in tellurite response irrespective of oxygen.

Gene enrichment analysis was performed on the sets of common negatively enriched and common positively enriched genes. We used DAVID and other bioinformatics resources to find gene ontology (GO) terms or KEGG pathways significantly overrepresented in these gene sets [75]. In particular, we noted which cellular processes were highlighted.

### 4.8. Metabolomic Profiling of Tellurite-Treated Cells

To investigate the metabolite-level changes caused by tellurite, we conducted an untargeted metabolomic analysis on *E. coli* cultures grown with and without tellurite in aerobic vs. anaerobic conditions. Cultures of *E. coli* BW25113 were grown in triplicate to OD_600_ ~0.5 in M9 minimal medium (0.2% glucose) under aerobic or anaerobic conditions. At this point, one set of cultures was treated with tellurite (5 µg/mL aerobic, 20 µg/mL anaerobic, analogous to earlier experiments) for 1 h, while control cultures received no tellurite. After 1 h, ~10 mL of each culture was rapidly centrifuged (4 °C, 5 min), and the cell pellet was quenched in liquid nitrogen. Metabolites were extracted using a cold biphasic solvent system optimized for broad metabolite coverage. Specifically, cell pellets were resuspended in 1:3:1 chloroform–methanol–water (vol/vol/vol) at 4 °C. The suspension was subjected to sonication to lyse cells, then incubated on ice for 10 min. After centrifugation (14,000× *g*, 10 min, 4 °C), the supernatant (polar + nonpolar metabolites) was collected and dried under vacuum. Dried extracts were resuspended in a small volume of 50% acetonitrile for analysis.

Metabolite analysis was performed using high-performance liquid chromatography coupled to mass spectrometry (HPLC-MS). We used a reverse-phase C18 column for separation. The LC gradient and MS settings were adjusted to detect a wide range of intracellular metabolites (including amino acids, organic acids, nucleotides, and lipids). MS data were acquired in both positive and negative electrospray ionization modes. Compound identification was achieved by matching *m*/*z* and retention time to a library of known *E. coli* metabolites (using standards and databases like KEGG). Where available, we also used fragmentation (MS/MS) to confirm metabolite identities.

Data processing was performed in MetaboAnalyst [76]. Detected features were log-transformed and normalized to internal standards. We performed univariate analysis to identify metabolites that showed significant changes (*p* < 0.05, Student’s *t*-test) upon tellurite treatment under both conditions. Fold changes [log_2_(FC)] of each metabolite in treated vs. control were calculated for aerobic and anaerobic samples. We then focused on metabolites whose abundance changed in response to tellurite in a similar way under both conditions (“common” metabolites). These included some that either decreased or increased under both conditions. Enrichment analysis of affected metabolites was conducted by mapping them to metabolic pathways (KEGG pathways).

### 4.9. Metabolite Supplementation Experiments

Based on the metabolomic findings, we hypothesized that the supplementation of certain depleted metabolites might alleviate tellurite toxicity, while adding those that accumulate might exacerbate it. To test this, we performed growth assays with added amino acids or nucleotide precursors in the presence of tellurite. *E. coli* BW25113 was grown in 96-well plates with M9 medium under aerobic or anaerobic conditions as before. We supplemented separate cultures with one of the following: L-methionine, L-leucine, L-phenylalanine, L-tryptophan, nucleotide pathway intermediates (AMP, ADP, ATP, GDP, UTP), or the cofactor pyridoxal 5′-phosphate (PLP, 0.1 mM). Each supplemented culture was then challenged with a range of tellurite concentrations (0, 0.5 and 5 µg/mL under aerobiosis; 0 and 5 µg/mL under anaerobiosis; etc., depending on the experiment. Controls without supplementation (i.e., cultures with or without tellurite, but without added amino acids) were included in all experiments to assess the baseline response. Growth was monitored, and %AUC was calculated as described above.

For amino acid supplementation, we particularly examined methionine, leucine, and phenylalanine because our metabolomics showed methionine tended to accumulate under tellurite stress, whereas leucine and phenylalanine levels dropped (especially under aerobic conditions). We measured how the presence of these amino acids (at 0.5 mM or 1 mM) influenced growth with sublethal tellurite. After 24 h, growth inhibition was assessed. Intracellular levels of the supplemented amino acids were quantified by UPLC-DAD after derivatization with DEEMM, following the method of Otaru et al. (2021) [77]. The samples were separated using a BEH C18 column (2.1 × 100 mm, 1.7 µm), and detection was carried out at 280 nm. This allowed us to confirm whether tellurite affected their uptake or utilization. All experiments were performed in at least triplicate. Two-way ANOVA was used to evaluate the interaction effects of supplementation and tellurite treatment on growth.

### 4.10. Statistical Analyses

Unless otherwise stated, data are reported as mean ± standard deviation of at least three independent experiments. Statistical significance between groups (e.g., treated vs. control, aerobic vs. anaerobic) was determined using unpaired two-tailed *t*-tests or ANOVA with appropriate post hoc tests for multiple comparisons. A threshold of *p* < 0.05 was considered significant. For the chemical-genomic screen, the significance of gene enrichment was determined by z-scores (as described in Section 4.7), and false-discovery rate (FDR) correction was applied to GO enrichment results. For metabolomics, an FDR-corrected *p* < 0.05 was used to identify significantly changed metabolites. Growth curve AUC comparisons used ANOVA and Tukey’s test. All statistical analyses were carried out using GraphPad Prism 7 or R software v0.3.0.

## 5. Conclusions

This study explores the oxidative and non-oxidative mechanisms by which tellurite exerts toxicity in *E. coli*. We found that *E. coli* is significantly more tolerant to tellurite under anaerobic conditions, primarily due to the reduction in reactive oxygen species (ROS)-mediated damage. However, through a combination of chemical-genomic profiling and metabolomic analyses, we discovered several toxic effects of tellurite that occur independently of oxygen. Tellurite disrupts both amino acid and nucleotide metabolism, leading to the accumulation of certain amino acids to inhibitory levels while depleting others that are essential for protein synthesis. It also affects enzymes with iron-sulfur cofactors and other metalloenzymes, disrupting the tricarboxylic acid (TCA) cycle and other metabolic pathways, regardless of the oxygen availability. Additionally, tellurite induces significant remodeling of the plasma membrane, notably increasing phospholipid methylation and fluidity. This alteration appears to be a bacterial adaptive response associated with enhanced tolerance in anaerobic environments.

Our findings indicate that tellurite toxicity is complex and involves multiple factors. In addition to generating oxidative stress, tellurite acts as a metabolic poison and a thiol-reactive agent. *E. coli* responds to this challenge through various mechanisms, triggering stress regulons, modifying its metabolism to prioritize survival over growth, and chemically reducing the toxin when possible.

These insights enhance our understanding of how bacteria manage chalcogen stress and could inform future approaches to using tellurite and similar compounds in antimicrobial or bioremediation strategies. By targeting the identified pathways—such as increasing thiol levels or addressing imbalances in amino acids—we may be able to improve bacterial resistance for biotechnological applications. Conversely, these strategies could also help increase the effectiveness of tellurite against pathogens in specific situations and environments.

## Figures and Tables

**Figure 1 ijms-26-07287-f001:**
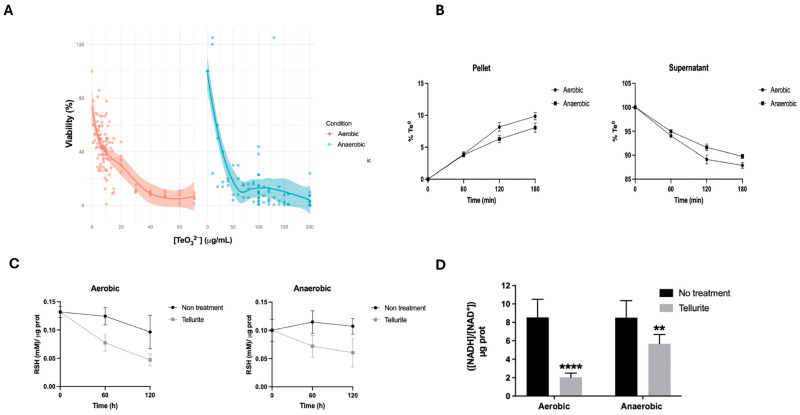
Resistance of *E. coli* to tellurite. Viability. *E. coli* BW25113 cultures in M9 medium at OD_600_ 0.5 were treated with different concentrations of tellurite for 1 h to quantify CFU/mL and estimate % viability (n = 9) (**A**). Tellurite quantification in *E. coli* cultures. Cultures grown in M9 medium at OD_600_ 0.5 were treated with 150 and 200 µg/mL of tellurite in aerobic (circle) and anaerobic (square) conditions, respectively (n = 3) (**B**). Effect of tellurite on thiol group (TGR) content in *E. coli* cultures. Cultures grown in M9 medium at OD_600_ 0.5 were treated with 5 and 20 µg/mL of tellurite in aerobic and anaerobic conditions, respectively (n = 5) (**C**). Energy status of *E. coli* treated with tellurite in aerobic and anaerobic conditions. Cultures grown in M9 OD_600_ 0.5 medium were treated for 1 h with 5 and 20 µg/mL of tellurite in aerobic and anaerobic conditions, respectively. The [NADH] and [NAD^+^] present in the cultures were then isolated and quantified. ** *p* = <0.01, **** *p* = <0.0001 (n = 6) (**D**).

**Figure 2 ijms-26-07287-f002:**
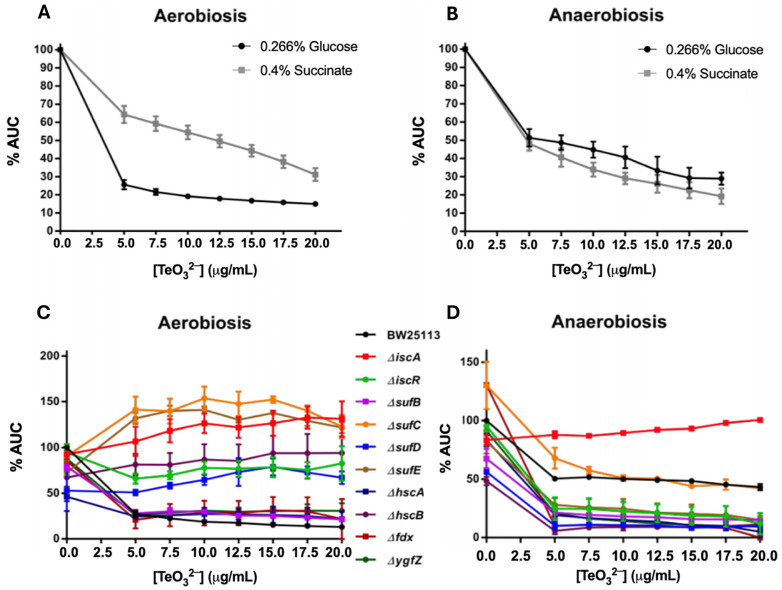
Growth of *E. coli* and mutants exposed to tellurite. Top panels, %AUC of growth curves of *E. coli* BW25113 in MOPS medium supplemented with different carbon sources and tellurite concentrations under (**A**) aerobiosis and (**B**) anaerobiosis; bottom panels, %AUC of growth curves of *E. coli* mutants in the presence of tellurite under (**C**) aerobiosis and (**D**) anaerobiosis (n = 7).

**Figure 3 ijms-26-07287-f003:**
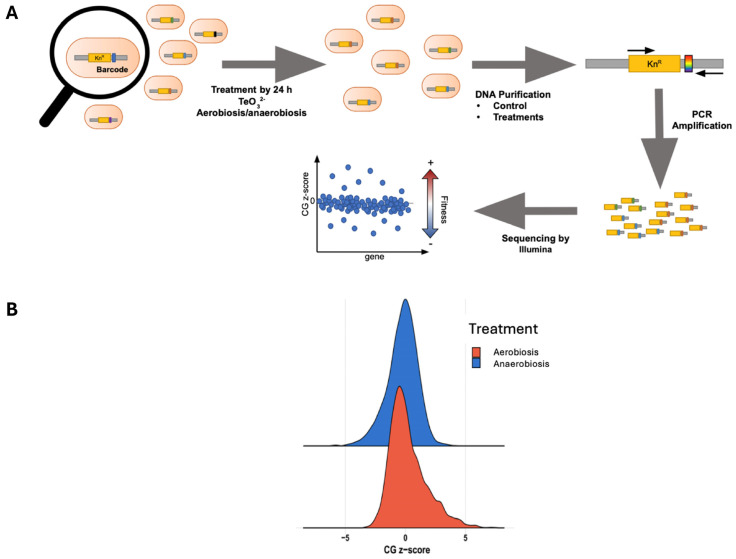
Genomic chemical profiling (GC). Simplified scheme of the experiment. The collection of mutants was exposed, under aerobic or anaerobic conditions, to a concentration of tellurite that produced 20% growth inhibition. After treatment, genomic DNA was purified from the collection, and the barcodes were amplified by PCR for subsequent sequencing and quantification by Illumina^®^. The CG z-score for each gene was calculated from these data (**A**). Distribution of CG z-scores under aerobiosis and anaerobiosis. Density plots show the distribution of the obtained CG z-scores. The data were normalized to standardized unit Z; thus, the mean of the distribution is 0, and the standard deviation is 1. The data corresponds to the distribution of the average CG z-scores from three biological replicates (**B**).

**Figure 4 ijms-26-07287-f004:**
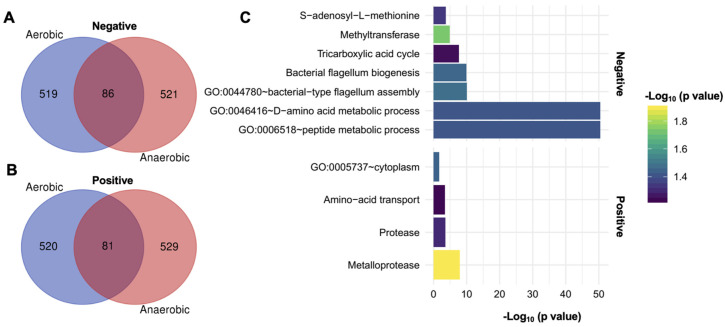
Gene enrichment analysis under aerobic and anaerobic conditions. Venn diagrams of genes common to aerobic and anaerobic conditions, (**A**) negatively enriched and (**B**) positively enriched. (**C**) Common enriched ontology terms from the CG analysis, *p*-value < 0.05.

**Figure 5 ijms-26-07287-f005:**
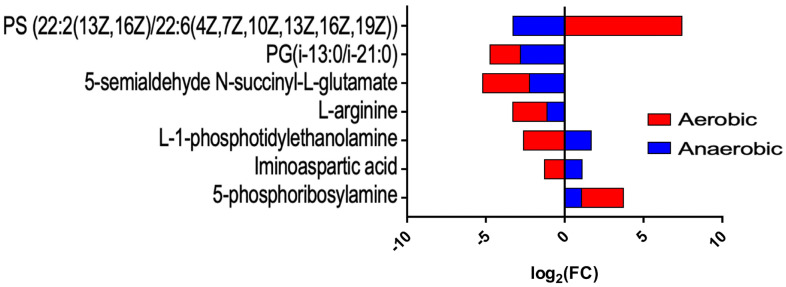
Enriched metabolites found in *E. coli* exposed to tellurite in aerobic and anaerobic conditions. Phosphatidylglycerol (PG), phosphatidylserine (PS) (n = 3).

**Figure 6 ijms-26-07287-f006:**
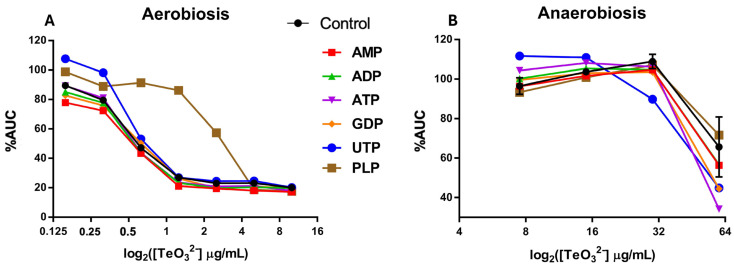
Effect of supplementation of nucleotide metabolism intermediates. %AUC of growth curves of E. coli BW25113 supplemented with 0.1 mM adenosine monophosphate (AMP), adenosine diphosphate (ADP), adenosine triphosphate (ATP), guanosine diphosphate (GDP), uridine triphosphate (UTP), or pyridoxal phosphate (PLP) in M9 medium in the presence of different concentrations of tellurite in (**A**) aerobiosis and (**B**) anaerobiosis (n = 3).

**Figure 7 ijms-26-07287-f007:**
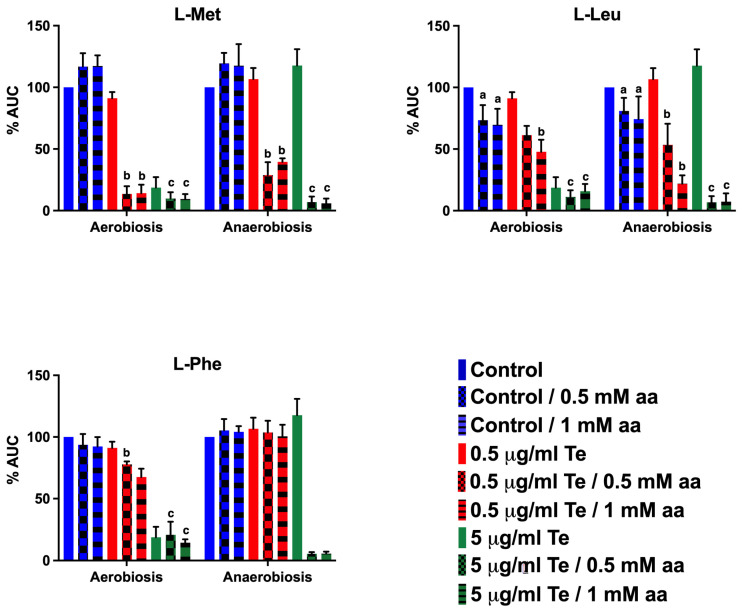
Effect of selected amino acids on tellurite toxicity in *E. coli* under aerobic and anaerobic conditions. The figure shows %AUC (area under the curve) values relative to untreated controls, for cultures grown in LB minimal medium with 0.2 mM methionine, leucine, or phenylalanine, with or without tellurite. Level of statistical significance relative to the corresponding treatment without amino acids. a, *p* < 0.001; b, *p* < 0.01; c, *p* < 0.05. Data for additional amino acids are presented in Appendix A.

**Figure 8 ijms-26-07287-f008:**
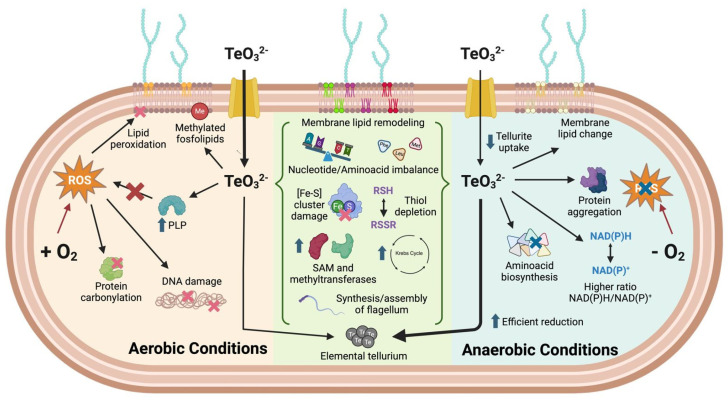
Schematic summary of tellurite toxicity mechanisms in *E. coli* under aerobic and anaerobic conditions. Under aerobic conditions, tellurite induces reactive oxygen species (ROS), leading to lipid peroxidation, protein carbonylation, and DNA damage. It also disrupts [Fe–S] enzymes and thiol homeostasis. Under anaerobic conditions, oxidative damage is diminished, but tellurite still alters amino acid biosynthesis, redox balance, and promotes protein aggregation. Common mechanisms in both conditions include nucleotide and amino acid imbalances, Fe–S cluster targeting, lipid remodeling, and tellurium reduction to elemental Te.

**Table 1 ijms-26-07287-t001:** Common compounds identified in tellurite treated *E. coli* under different oxygen conditions.

Compounds	KEGG IDs	Aerobic		Anaerobic		Metabolic Pathway
		log_2_(FC)	log_10_(p)	log_2_(FC)	log_10_(p)	
5-phosphoribosylamine	C03090	2.659	1.854	1.071	1.865	Biosynthesis of 5-aminoimidazole ribonucleotides I and II
Iminoaspartic acid	C05840	−1.273	1.438	1.107	1.564	NAD^+^ de novo biosynthesis I (from aspartate)
L-1-phosphatidylethanolamine	C00350	−2.609	1.329	1.679	1.253	Biosynthesis of cardiolipin III, PS and PE
L-arginine	C00062	−2.164	1.507	−1.120	1.116	Degradation of L-arginine III, biosynthesis of putrescine
5-semialdehyde N-succinyl-L-glutamate	C05932	−2.965	1.204	−1.226	1.440	Degradation of L-arginine II
PG(i-13:0/i-21:0)	n/a	−1.918	2.045	−2.811	1.392	Lipid metabolism
PS(22:2(13Z,16Z)/22:6(4Z,7Z,10Z,13Z,16Z,19Z))	n/a	7.438	2.266	−3.277	1.147	Lipid metabolism

Phosphatidylglycerol (PG), phosphatidylserine (PS).

**Table 2 ijms-26-07287-t002:** Compounds found in *E. coli* treated with tellurite under different oxygen conditions.

		Compound	KEGG IDs	log2(FC)	log10(p)	Metabolic Pathway
Aerobic	Negative Enrichment	PG(16:0/20:3(8Z,11Z,14Z))	n/a	−2.149	1.693	Lipid metabolism
		PG(i-13:0/i-19:0)	n/a	−6.983	1.612	Lipid metabolism
		PGP(22:6(4Z,7Z,10Z,13Z,16, 19Z)/20:2(11Z,14Z))	n/a	−1.668	1.526	Lipid metabolism
		L-tryptophan	C00078	−1.164	1.124	Biosynthesis and degradation II (pyruvate pathway) of L-tryptophan
		S-lactoylglutathione	C03451	−1.295	3.112	Degradation I of methylglyoxal
		Adenosine phosphosulfate	C00224	−1.573	2.850	Sulfate activation for sulfonation
		Acetyl-CoA	C00024	−4.385	2.208	Krebs cycle I, biosynthesis of fatty acids, L-cysteine and L-leucine
	Positive Enrichment	PE-NMe(16:1(9Z)/18:0)	n/a	2.771	2.289	Lipid metabolism
		PE-NMe2(14:0/16:1(9Z))	n/a	7.343	1.711	Lipid metabolism
		PE-NMe2(16:1(9Z)/16:1(9Z))	n/a	6.034	1.214	Lipid metabolism
		PS(18:0/20:3(8Z,11Z,14Z))	n/a	2.557	1.870	Lipid metabolism
		PS(22:5(7Z,10Z,13Z,16Z,19Z)/20:2(11Z,14Z))	n/a	1.251	1.146	Lipid metabolism
Anaerobic	Negative Enrichment	PA(20:5(5Z,8Z,11Z,14Z,17Z)/24:1(15Z))	n/a	−3.192	2.158	Lipid metabolism
		PA(24:1(15Z)/18:0)	n/a	−1.694	2.195	Lipid metabolism
		PG(a-13:0/i-24:0)	n/a	−2.014	1.543	Lipid metabolism
		PS(22:4(7Z,10Z,13Z,16Z)/22:5(7Z,10Z,13Z,16Z,19Z))	n/a	−3.655	1.743	Lipid metabolism
		2′-deoxycytidine 5′-monophosphate (dCMP)	C00239	−1.860	1.100	Phosphorylation of pyrimidine deoxyribonucleotides
	Positive Enrichment	glycerol 2-phosphate	C02979	1.000	1.162	Glycerol and phosphate precursor
		PA(20:4(5Z,8Z,11Z,14Z)/24:1(15Z))	n/a	1.679	1.173	Lipid metabolism
		PA(22:4(7Z,10Z,13Z,16Z)/20:0)	n/a	1.786	1.754	Lipid metabolism
		PE(18:4(6Z,9Z,12Z,15Z)/P-18:1(9Z))	n/a	2.158	1.147	Lipid metabolism
		PG(a-13:0/i-22:0)	n/a	1011	1155	Lipid metabolism
		L-dihydroorotic acid	C00337	1.497	1.746	Biosynthesis of UMP I

Phosphatidic acid (PA), phosphatidylethanolamine (PE), phosphatidylglycerol (PG), phosphatidylglycerol phosphate (PGP), and phosphatidylserine (PS).

**Table 3 ijms-26-07287-t003:** Effect of tellurite on the intracellular concentration of amino acids.

Amino Acid	Aerobiosis			Anaerobiosis		
µM/mg Protein	Control	Tellurite	*p*	Control	Tellurite	*p*
Alanine	457.8 ± 48.1	687.6 ± 49.2	ns	1229.9 ± 147.6	1223.8 ± 211.2	ns
Asparagine	171.8 ± 6.3	16.2 ± 3.1	ns	90.4 ± 36.7	63.1 ± 12.3	ns
Aspartic Acid	4503.7 ± 188.6	440.2 ± 54.7	***	1793.2 ± 764.4	2111.3 ± 526.7	ns
Glutamic Acid	1201.5 ± 44.5	1278.5 ± 151.1	ns	2607.7 ± 292.7	2257.7 ± 63.2	ns
Glutamine	47.7 ± 10.0	15.5 ± 3.0	ns	180.8 ± 10.2	105.1 ± 23.8	**
Glycine	156.2 ± 4.1	63.0 ± 5.1	****	128.5 ± 8.4	127.6 ± 3.9	ns
Histidine	22.2 ± 0.7	10.4 ± 2.1	ns	22.8 ± 7.3	14.5 ± 4.4	ns
Isoleucine	5.7 ± 2.0	2.3 ± 0.8	*	7.0 ± 1.7	7.5 ± 0.5	ns
Lysine	188.3 ± 5.1	364.9 ± 104.2	*	175.8 ± 16.1	146.5 ± 40.3	ns
Methionine	9.5 ± 4.4	16.0 ± 0.5	*	11.9 ± 1.7	12.6 ± 0.4	ns
Phenylalanine/Leucine	18.4 ± 3.0	28.3 ± 0.6	*	19.0 ± 5.7	16.8 ± 3.1	ns
Proline	4.7 ± 2.0	9.2 ± 1.1	ns	49.8 ± 2.6	67.4 ± 11.0	*
Serine	21.3 ± 1.4	10.8 ± 1.7	***	9.0 ± 0.8	8.6 ± 0.9	ns
Threonine	500.4 ± 66.7	36.0 ± 4.8	****	114.3 ± 14.0	96.1 ± 16.2	ns
Tyrosine	28.7 ± 9.4	99.2 ± 2.8	**	58.0 ± 8.0	61.0 ± 61.0	ns
Valine	15.1 ± 0.1	21.4 ± 2.1	*	15.0 ± 2.5	18.5 ± 1.8	ns

ns non significative; * *p* < 0.05; ** *p* < 0.01; *** *p* < 0.001; **** *p* < 0.0001.

## Data Availability

All data generated or analyzed during this study are included in this published article and the Appendix A.

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
