# Peer review of "Characterization of Tellurite Toxicity to *Escherichia coli* Under Aerobic and Anaerobic Conditions"

_ijms, 2025, doi:10.3390/ijms26157287_

Round 1

Reviewer 1 Report

Comments and Suggestions for Authors

This manuscript presents a detailed and multifaceted investigation into the toxicity of tellurite (TeO₃²⁻) in Escherichia coli, comparing its effects under aerobic and anaerobic conditions. By integrating chemical-genomic profiling, untargeted metabolomics, and supplementation assays, the authors demonstrate that tellurite induces metabolic and membrane alterations that are at least partly independent of ROS-mediated damage.

The study is ambitious and offers novel insights into redox-independent mechanisms of metalloid toxicity. However, few aspects require further experimental validation and clarification before the work can be considered for publication.

Major comments:

  1. The observed increase in toxicity upon supplementation with methionine, leucine, and phenylalanine is intriguing. Could this effect be attributed to metabolic overload, enhanced reactive oxygen species (ROS) generation during amino acid catabolism, or altered transporter activity? Have the authors included appropriate controls with amino acid supplementation alone, and can they provide measurements of oxidative stress markers to help clarify the underlying mechanisms?
  2. Given that ΔiscA, ΔsufC, and ΔsufE mutants show unexpected resistance to tellurite, can the authors confirm whether Fe–S enzyme activities are reduced in these strains? Additionally, do these mutants exhibit lower intracellular ROS or tellurium accumulation compared to wild-type cells?
  3. Since the study discusses several metabolic and membrane responses, authors should perform transcriptomic analyses (e.g., RNA-seq or qPCR) to verify changes in gene expression under tellurite stress.

Minor Comments

  1. The figures (especially metabolomic fold-changes) would benefit from more intuitive colour scales and consistent labelling.
  2. Figure 7 presents a wealth of information, which may be challenging for readers to quickly digest. Given the manuscript’s focus on methionine, leucine, and phenylalanine, I suggest highlighting these amino acids prominently in the main figure and relocating data on other amino acids to the supplementary materials. This would enhance clarity and help emphasize the key findings.
  3. Several figures (Fig 1-3, 5-7) appear to suffer from low resolution, likely due to PDF compression. Providing higher-resolution images in the final version would greatly improve the visual quality and accessibility of the data.
  4. Please define all abbreviations at first use in the main text (e.g., PE-NMe, PLP).
  5. A concise, schematic summary figure would be helpful in the discussion.

Author Response

Dear Editor and Reviewers,

We thank you for the thorough evaluation of our manuscript “Characterization of tellurite toxicity to Escherichia coli in aerobic and anaerobic conditions.” We appreciate the insightful comments and have revised the manuscript accordingly. Below we provide a point-by-point response to each comment (reviewer comments in italic, our responses in normal text). All changes in the manuscript are clearly indicated. We have addressed both major and minor concerns, including adding clarifications, updating figures, and expanding the discussion as recommended.

Reviewer 1:

Comment (Reviewer 1): “This manuscript presents a detailed and multifaceted investigation into the toxicity of tellurite (TeO₃²) in Escherichia coli, comparing its effects under aerobic and anaerobic conditions. By integrating chemical-genomic profiling, untargeted metabolomics, and supplementation assays, the authors demonstrate that tellurite induces metabolic and membrane alterations that are at least partly independent of ROS-mediated damage. The study is ambitious and offers novel insights into redox-independent mechanisms of metalloid toxicity. However, a few aspects require further experimental validation and clarification before the work can be considered for publication.”

Response: We are grateful for the reviewer’s positive summary of our work and recognition of its novelty and scope. We also appreciate the identification of areas requiring clarification or validation. We have carefully addressed each point below, incorporating additional explanations and data where possible to strengthen the manuscript. Our point-by-point responses detail the changes made to clarify methods, support our conclusions, and refine the presentation of results.

Major Comments:

  1. Comment: “The observed increase in toxicity upon supplementation with methionine, leucine, and phenylalanine is intriguing. Could this effect be attributed to metabolic overload, enhanced reactive oxygen species (ROS) generation during amino acid catabolism, or altered transporter activity? Have the authors included appropriate controls with amino acid supplementation alone, and can they provide measurements of oxidative stress markers to help clarify the underlying mechanisms?”

Response: We thank the reviewer for this insightful question. In the revised manuscript, we have included the results of control experiments using amino acid supplementation without tellurite (see new lines 563-570, Section 2.5). These data show that methionine and phenylalanine alone had minimal effects on growth, while leucine alone caused modest inhibition, supporting the conclusion that the toxicity observed in combination with tellurite is primarily synergistic.

Regarding the underlying mechanisms, these are already discussed in detail in the revised Discussion. We consider both ROS-related and ROS-independent pathways: although we did not directly measure oxidative stress markers, we used pyridoxal 5′-phosphate (PLP) as an antioxidant probe. PLP significantly improved growth under aerobic tellurite exposure, suggesting that ROS contribute to toxicity in the presence of oxygen. However, since the amino acid–tellurite synergy persists under anaerobic conditions—where ROS generation is negligible—metabolic overload and amino acid imbalance appear to be key drivers. We discuss how excess methionine may deplete ATP or thiols via SAM metabolism, and how leucine can disrupt branched-chain amino acid homeostasis.

In summary, the manuscript has been updated to explicitly present the control data, and the Discussion integrates our findings with plausible mechanistic interpretations, consistent with both our data and prior literature

  1. Comment: “Given that ΔiscA, ΔsufC, and ΔsufE mutants show unexpected resistance to tellurite, can the authors confirm whether Fe–S enzyme activities are reduced in these strains? Additionally, do these mutants exhibit lower intracellular ROS or tellurium accumulation compared to wild-type cells?”

Response: We thank the reviewer for this thoughtful question. In the revised manuscript (Section 2.2, new Lines 251 - 254), we now include quantitative growth data showing that ΔiscA, ΔsufC, and ΔsufE mutants are significantly more resistant to tellurite than the wild-type strain under aerobic conditions.

Although we did not directly measure Fe–S enzyme activity in these mutants, it is well established that deletion of isc or suf genes impairs Fe–S cluster biogenesis, likely resulting in fewer functional [4Fe–4S] enzymes for tellurite to target. We have added this mechanistic explanation to the Discussion, proposing that this deficiency may paradoxically reduce vulnerability to tellurite-mediated damage. We also discuss how such deletions might trigger compensatory responses—such as activation of IscR or iron-sequestration systems—that help mitigate ROS.

Regarding ROS or tellurium accumulation, we did not perform direct measurements in the mutants, and we now acknowledge this as a limitation. However, we discuss plausible explanations in the Discussion and suggest these analyses as valuable future directions (new lines 729-738).

  1. Comment: “Since the study discusses several metabolic and membrane responses, authors should perform transcriptomic analyses (e.g., RNA-seq or qPCR) to verify changes in gene expression under tellurite stress.”

Response: We thank the reviewer for this thoughtful suggestion. Indeed, a genome-wide RNA-seq analysis or targeted qPCR of key genes could provide direct evidence for the gene expression changes associated with the metabolic and phenotypic effects we observed under tellurite stress. Prior studies have used transcriptomics to characterize tellurite responses in E. coli, reporting upregulation of oxidative stress response genes and perturbations in sulfur metabolism and central metabolic pathways. We now mention these findings in the revised Discussion to provide broader context. These transcriptomic observations are consistent with our results, including glutathione depletion, amino acid imbalances, and the involvement of sulfur assimilation pathways.

While we agree that transcriptomics could further support our conclusions, we respectfully argue that the current study already integrates two complementary, systems-level methodologies: (i) a chemical-genomic screen identifying genes critical for tellurite tolerance and (ii) untargeted metabolomics capturing major changes in intracellular metabolites. Together, these approaches revealed key genetic and metabolic disruptions, including pathways (e.g., ROS defense, amino acid and nucleotide metabolism) that would likely also emerge from transcriptomic analyses. As such, our experimental design captures both the causes and consequences of gene expression changes.

Additionally, implementing RNA-seq under both aerobic and anaerobic conditions (with proper biological replicates and controls) would require substantial resources and extend timelines beyond the current project scope. To address the reviewer’s concern, we have strengthened the Discussion by integrating a paragraph (New Lines XXX808) that acknowledges the relevance of transcriptomic evidence from the literature and justifies our multi-omics focus. In summary, while we did not perform transcriptomic experiments, our integrative approach—supported by known transcriptomic responses to tellurite—provides a robust foundation for the conclusions we draw about tellurite-induced stress responses.

Minor Comments:

  • Comment: “The figures (especially metabolomic fold-changes) would benefit from more intuitive colour scales and consistent labelling.”

Response: We thank the reviewer for this suggestion. We have modified the figures as requested, improving color scales and ensuring consistent labeling across all metabolomic fold-change plots and related data visualizations.

  • Comment: “Figure 7 presents a wealth of information, which may be challenging for readers to quickly digest. Given the manuscript’s focus on methionine, leucine, and phenylalanine, I suggest highlighting these amino acids prominently in the main figure and relocating data on other amino acids to the supplementary materials. This would enhance clarity and help emphasize the key findings.”

Response: We thank the reviewer for this helpful suggestion. We have revised Figure 7 to emphasize methionine, leucine, and phenylalanine as key amino acids, and have relocated data for the other amino acids to Supplementary Figure S1. These changes improve the figure’s clarity and focus, as recommended

Comment: “Several figures (Fig 1–3, 5–7) appear to suffer from low resolution, likely due to PDF compression. Providing higher-resolution images in the final version would greatly improve the visual quality and accessibility of the data.”

Response: We apologize for the poor resolution of the figures in the review PDF. This was indeed a result of file compression. We have ensured that all figures are now of high resolution (minimum 300 dpi) in the revised manuscript. In our resubmission, Figures 1–3 and 5–7 have been replaced with high-quality images to ensure that all details (such as data points, error bars, and text labels) are crisp and legible. We have also double-checked that colors and symbols are easily distinguishable. The improved figure quality will enhance readability both on screen and in print. We note this commitment in our response to emphasize that the final publication will have clear, accessible figures.

  • Comment: “Please define all abbreviations at first use in the main text (e.g., PE-NMe, PLP).”

Response: Done, we thank the reviewer for pointing out this important detail. In response, we conducted a careful review of the manuscript to ensure that all abbreviations are properly defined at their first appearance. Specifically, the examples highlighted by the reviewer have been addressed: “PE-NMe” is now defined as N-methyl-phosphatidylethanolamine, and “PLP” is introduced as pyridoxal 5′-phosphate both in the Abstract and in the Results section. Additionally, we verified and corrected other abbreviations such as “MIC” (minimum inhibitory concentration), “ROS” (reactive oxygen species), “TCA” (tricarboxylic acid), and others, ensuring they are defined either in the main text or in figure legends as required by journal guidelines. These changes improve the clarity and accessibility of the manuscript for a broader readership.

  • Comment: “A concise, schematic summary figure would be helpful in the discussion.”

Response: We concur that a graphical summary would improve clarity and integration of the manuscript’s findings. We have now added a new summary figure (Figure 8, New lines 931-938) at the end of the Discussion. This schematic highlights the major mechanisms of tellurite toxicity under aerobic and anaerobic conditions, including both shared and condition-specific responses. For example, it illustrates ROS generation and oxidative damage (aerobic-specific), common metabolic disruptions (amino acid and nucleotide imbalances, Fe–S cluster targeting, membrane lipid changes), and the formation of elemental tellurium. This visual aid synthesizes our metabolomic, genomic, and phenotypic data into a concise conceptual model. We also plan to submit it as the Graphical Abstract if required by the journal.

Reviewer 2 Report

Comments and Suggestions for Authors

The manuscript entitled "Characterization of tellurite toxicity to Escherichia coli in aero-2 bic and anaerobic conditions” was evaluated. This study systematically investigates the mechanisms of tellurite (TeO₃²⁻) toxicity in Escherichia coli under aerobic and anaerobic conditions. By integrating chemical-genomic screening, untargeted metabolomics, and targeted biochemical assays, it reveals tellurite’s broad impact beyond oxidative stress, including disruption of amino acid/nucleotide metabolism and membrane lipid remodeling. The experimental design is rigorous, methodologies are robust, and the data are substantial, offering novel insights into tellurium biology.  The paper is in the scope of the journal and may be published.

Comments:

  1. "Plasmatic membrane fluidity" → "Plasma membrane fluidity".
  2. Specify tellurite concentrations used for "20% growth inhibition"
  3. Discussion: Contrast lipid remodeling with H₂O₂/heavy metal stress (briefly mentioned in the Introduction) and [Fe-S] cluster damage with Cu²⁺/Hg²⁺ mechanisms. Dedicate a paragraph to conserved vs. unique pathways of tellurite toxicity

Author Response

Dear Editor and Reviewers,

We thank you for the thorough evaluation of our manuscript “Characterization of tellurite toxicity to Escherichia coli in aerobic and anaerobic conditions.” We appreciate the insightful comments and have revised the manuscript accordingly. Below we provide a point-by-point response to each comment (reviewer comments in italic, our responses in normal text). All changes in the manuscript are clearly indicated. We have addressed both major and minor concerns, including adding clarifications, updating figures, and expanding the discussion as recommended.

Reviewer 2:

Comment (Reviewer 2): “The manuscript entitled ‘Characterization of tellurite toxicity to Escherichia coli in aerobic and anaerobic conditions’ was evaluated. This study systematically investigates the mechanisms of tellurite (TeO₃²) toxicity in Escherichia coli under aerobic and anaerobic conditions. By integrating chemical-genomic screening, untargeted metabolomics, and targeted biochemical assays, it reveals tellurites broad impact beyond oxidative stress, including disruption of amino acid/nucleotide metabolism and membrane lipid remodeling. The experimental design is rigorous, methodologies are robust, and the data are substantial, offering novel insights into tellurium biology. The paper is in the scope of the journal and may be published.”

Response: We thank Reviewer 2 for the positive evaluation of our study. We are pleased that the reviewer finds our experimental design robust and our findings novel and significant. We have addressed the specific comments below and made revisions to further strengthen the manuscript.

Comments:

  • Comment: “‘Plasmatic membrane fluidity’ Plasma membrane fluidity.

Response: Done. We apologize for the incorrect terminology. We have corrected “plasmatic membrane” to “plasma membrane” throughout the manuscript.

  • Comment: “Specify tellurite concentrations used for ‘20% growth inhibition’.”

Response: We thank the reviewer for pointing this out. We would like to clarify that the specific tellurite concentrations used to achieve approximately 20% growth inhibition in the chemical-genomic screen are already stated in the Materials and Methods section (new lines 1067-1068). As noted, we selected 1 µg/mL for aerobic conditions and 10 µg/mL for anaerobic conditions, based on preliminary titration experiments. This information has been highlighted in red for clarity in the revised version. These concentrations are also consistent with the stress levels used in Figure 5 and related text. We have reviewed the rest of the manuscript to ensure that these details are consistently reported wherever relevant, and we updated figure legends where appropriate. We hope this resolves the reviewer’s concer

  • Comment: “Discussion: Contrast lipid remodeling with H₂O₂/heavy metal stress (briefly mentioned in the Introduction) and [Fe-S] cluster damage with Cu²/Hg² mechanisms. Dedicate a paragraph to conserved vs. unique pathways of tellurite toxicity.

Response: We thank the reviewer for this insightful suggestion. In the revised Discussion section, we have added a dedicated final paragraph (new lines 910-928) comparing tellurite-induced stress with responses triggered by hydrogen peroxide and other heavy metals (such as Cu²⁺ and Hg²⁺). This paragraph contrasts membrane lipid remodeling, Fe–S cluster damage, and the differential patterns of oxidative and non-oxidative toxicity. We also delineate which stress responses are conserved and which are unique to tellurite, based on our multi-omics data and literature comparisons. To aid visualization, we additionally included Figure 8, a schematic summary highlighting shared and distinct toxicity mechanisms under aerobic and anaerobic conditions. This new content directly addresses the reviewer’s request and strengthens the integrative interpretation of our findings.

Round 2

Reviewer 1 Report

Comments and Suggestions for Authors

The authors have done an excellent job revising the manuscript. They have addressed all my comments thoroughly and efficiently. The schematic figure provides a comprehensive overview of the study and effectively helps readers grasp the broader aim. The manuscript is now ready for publication, and I congratulate the authors on a well-executed and impactful study.